# Numerical Evaluation of the Fracture Process of 41Cr4 Steel: Analysis of Cracks Grown in a Plane Strain State Domination Based on Experimental Results

**DOI:** 10.3390/ma15207361

**Published:** 2022-10-20

**Authors:** Marcin Graba

**Affiliations:** Department of Metrology and Unconventional Manufacturing Methods, Faculty of Mechatronics and Mechanical Engineering, Kielce University of Technology, 25-314 Kielce, Poland; mgraba@tu.kielce.pl

**Keywords:** 41Cr4 steel, FEM, *J*-integral, CTOD, crack initiation moment, stress distribution, geometric constraints, plastic zone, crack growth

## Abstract

The paper presents the evaluation of the fracture process of 41Cr4 steel, based on experimental tests and numerical analysis carried out for growing cracks. Selected results of experimental studies presented in previous publications in 2015–2016 which were used in the construction of FEM models of SEN(B) specimens dominated by a plane strain state, are quoted. Using FEM simulations, the tensile curves and crack growth curves recorded during laboratory tests, several calculations were performed, among which can be distinguished the evaluation of plastic zones in the considered specimens, stress distributions, selected measures of geometric constraints, accumulated plastic strains and selected parameters of fracture mechanics. The numerical calculations presented in this paper allow to estimate the changes of the *J*-integral for the tested steel, the crack tip opening displacement, the size of the plastic zone and the stress distribution in front of the crack tip. Comprehensive numerical analysis with the use of the actual tensile curve clearly characterizes the behavior of the material under the influence of increasing external loads.

## 1. Introduction 

For many years, the problem of determining the fracture toughness of construction materials has appeared in the field of fracture mechanics. This concerns both elastic and elastic–plastic problems. Depending on the type of problem, or rather the material, which the engineer has to deal with while solving many structural and strength problems, appropriate measures of fracture toughness should be used, among which, the critical values of the stress intensity factor (SIF), *K_C_*, the *J*-integral, denoted as *J_C_*, and the crack tip opening displacement, *δ_C_*, determined in the laboratory according to strictly defined recommendations [1,2,3], can be considered as material features and proper fracture toughness, denoting these values as *K_IC_*, *J_IC_* or *δ_IC_*. For many years, these values were considered as material constants, however, the scientific discussions in the field of fracture mechanics suggest that these values should be considered as material features rather than material constants [4].

The standards [1,2,3] are perfect for determining the fracture toughness of materials known to break, as well as thermochemically treated materials. However, in [5], it was shown that in the case of a material of blind origin—only the designation of the material was known—41Cr4 steel (formerly 40H steel according to the Polish Standard), an engineer seeking to determine the fracture toughness will encounter many problems that will not allow him to clearly define these features of the material. The analysis carried out in the study, originally according to the world-renowned ASTM standard [1], did not allow us to estimate the fracture toughness according to these recommendations, due to the lack of valid points [5]. One of the solutions was to use the standard still in force in Poland [2], which led to quite divergent results [5,6]. In addition, in [6], in which the results given in [5] were quoted and numerical calculations carried out for stationary fractures, assuming small and large deformations, were supplemented, an attempt was made to estimate the conventional moment of crack initiation, based on the ASTM standard [1], relating its fragment to the determination of the *K_IC_* fracture toughness for brittle materials. However, this approach also led to heterogeneous results [5,6].

The material analyzed in the papers [5,6]—41Cr4 steel, previously designated as 40H—was obtained directly from the actual structure. All material constants were determined during uniaxial tensile tests, using specimens with a rectangular cross-section [5,6]. The conducted tests showed that the tested material is characterized by a clear yield point, a significant total elongation and an average hardening exponent *n* in the Ramberg–Osgood law of less than 9 [5,6]. The tests carried out in the scope of determining the fracture toughness of elastic–plastic materials, *J_C_*, which were made with the use of SEN(B) three-point bending specimens with a width of *W* = 25 mm and a relative initial crack length *a_0_*/*W* = 0.50 [5,6], indicated the occurrence of significant deformations of plastic around the crack tip and the uneven, stepwise increase in the crack length [5,6]. All these tests made it possible to determine the *J*-*R* curves, presenting changes in the value of the *J*-integral as a function of crack length increments; however, these curves had an inappropriate number of valid points to be considered reliable according to the ASTM [1] standard [5,6]. The use of the obtained *J*-*R* curves in the procedure of determining the critical values of the *J*-integral, denoted as *J_C_* according to the Polish Standard [3], allows the implementation of the entire analysis algorithm and even, in selected cases, the determined value of *J_C_* can be considered a material feature, denoted as *J_IC_* [5,6].

Figure 1 presents selected results of experimental works obtained for 41Cr4 steel: tensile curve, selected recorded force diagrams as a function of crack mouth displacement—*P* = *f*(*δ_M_*), and the *J*-*R* curves estimated on their basis—*J* = *f*(*da*) diagrams, which were determined using the SEN(B) specimens recommended by the ASTM [1] standard (Figure 2). Table 1 presents extracts from the tabular data provided in [5,6], which relate to experimentally determined constants material and physical quantities, leading to an attempt to determine the fracture toughness and the conventional fracture initiation moment [5,6]. The three most interesting diagrams, *P* = *f*(*δ_M_*), were selected to present the behavior of the material that was tested (see Figure 1b) to illustrate the abrupt cracking of 41Cr4 steel, which is also shown in Figure 1c.

The material mentioned above—41Cr4 steel—is a commonly used material in Poland for the construction of various types of technical and structural facilities, which are used and operated today. Very often, when servicing these objects and assessing their physical condition, it turns out that they have very scarce design documentation, containing data on the material or geometric dimensions. An example of such a structure can be the fragment of the inroad shown in Figure 2a [5].

The inroad was destroyed during operation, and the resulting scrap—the frame material—was used to prepare the pucks for experimental tests. The presented structure was welded from bars with a square cross-section of 25 × 25 mm, and the average characteristic dimension of the weld is about 7–10 mm. After the destruction of the inroad, a significant part of the presented structure was cut into smaller pieces, and the fragments directly related to the welds and heat-affected zones were removed to prepare the samples for the static tensile test and fracture toughness tests.

Determination of the chemical composition allowed to state that the steel used for this structure corresponds to the composition of 41Cr4 steel, long marked as 40H in Poland. It is a very difficult to weld steel—it should be heated before welding and then heat-treated. It is usually used for heavily loaded shafts, connecting rods, bushings, axles, gears, abrasive discs, instrument bodies and longer-lasting molds. It is a chrome steel, intended for thermal improvement, with medium hardenability. Very often, in Poland, this steel was used in its raw state to build simple structures, which did not always turn out to be an appropriate solution, an example of which is the raid mentioned in [5].

The behavior of the material—41Cr4 steel, briefly discussed above—was presented in two studies, which have already been mentioned. One of them dealt with the subject of problems that arise during experimental studies in the field of the evaluation of fracture toughness, *J_C_* [5], and the other one extended the analysis of the obtained experimental results with numerical calculations carried out for stationary cracks [6]. These papers provide at least a partial answer on how to proceed when there are problems in the assessment of fracture toughness, as well as what may be the reasons for this, or how to interpret such behavior of the material, along with the assessment of various parameters and quantities considered important in the field of fracture mechanics.

To determine the material constants characteristic for the uniaxial tensile test, in accordance with the recommendations of the relevant standard, flat specimens were prepared with a rectangular cross-section, a_0_ × b_0_ = 2 × 10 mm (the active cross-sectional area was about S_0_ = 20 mm^2^), and the measurement base l_0_ = 50 mm (Figure 2a). However, to determine the fracture toughness, in accordance with the standards [1,2,3], it was decided to use the SEN(B) specimens: “single edge-notched specimen under bending”, a beam with one-sided fracture subjected to bending (Figure 2b). The SEN(B) specimens used in the experiments were characterized by a width of W = 25 mm, which determined the support spacing S = 4·W = 100 mm. When planning the tests, it was decided to prepare specimens with three thicknesses, B = {5, 10, 15} mm, and the relative length of the crack a/W = 0.50—the relative length of the crack is determined by the assumptions of the standards [1,2,3]. During the tests, 5 specimens were used in the scope of the static tensile test and 9 specimens in the scope of determining the fracture toughness. All strength tests and fracture toughness tests were carried out on the MTS 810 hydropulsion testing machine.

In [5,6], it was suggested that the value of the fracture toughness, *J_C_*, is determined for two specimens with a thickness of *B* = 15 mm, as appropriate. The average value is *J_IC_* = 75.5 N/mm, and in the case of numerical analysis for stationary fractures, it corresponds to the external load *P* fulfilling the condition *P*/*P_0_* = 1.19, as well as the maximum level of stress opening the crack surfaces, *σ_zz_*/*σ*_0_ = 4.37 (which occurs at the normalized distance from the crack tip *ψ* = 0.56, which corresponds to the physical distance *r* = 0.10 mm—this is 1/125·*b*, where *b* is the length of the non-cracked section of the specimen, calculated as *b* = *W* − *a*) [6]. Reference [6] presents a comprehensive analysis of various values of elasto-plastic fracture mechanics, which were numerically estimated for the case of stationary cracks, assuming large deformations, which are confirmed by significant plastic deformations observed during experimental tests of 41Cr4 steel [6]. However, it seems that this is not a sufficient assessment and analysis of the material behavior because of the large plastic deformations and step increments of the crack length, which nevertheless are significant and affect the form of the *P* = *f*(*δ_M_*) diagrams. A natural supplement to the already presented results [5,6] should be a numerical analysis of the crack growth, based on a properly constructed numerical model, considering the actual experimental tensile curve, as well as the actual *J*-*R* curve (the crack growth curve, presenting changes in the value of the *J*-integral as a function of crack length increments, *da*, which can be written as *J* = *f*(*da*)). For this reason, it was decided to adapt the three *J*-*R* curves presented in Figure 1c to the appropriate numerical model and, after performing numerical calculations for the case of increasing cracks, to analyze selected fracture mechanics parameters, which will be discussed in the following sections of this paper.

## 2. Details of Numerical Modeling for the Case of Growing Cracks

Numerical calculations of SEN(B) specimens made of 41Cr4 steel for the case of growing cracks, similarly to the case in the first section of the paper (stationary fracture analysis), were performed with the ADINA SYSTEM 8.8 package [7,8]. All numerical calculations were carried out for the actual material model (see Figure 1a), based on the actual tensile curve constituting the lower boundary of all tensile curves recorded during the experimental tests of 41Cr4 steel. The calculations assume the yield point of *σ_0_* = 439 MPa, Young’s modulus *E* = 209 GPa and Poisson’s coefficient *ν* = 0.3. During the FEM calculations, a homogeneous, isotropic, elastic–plastic model of the material was assumed. The Huber–Misses–Hencky yield condition was used in the calculations. The tensile curve, until reaching the yield point, was modeled as a straight line (strains were estimated based on their relationship with stresses using Hook’s law) [6]. After reaching the yield point, the coordinates of successive points of the model stretching curve were developed based on the lower boundary of the experimentally recorded stretching curves, using at the same time the recommendations of the authors of the ADINA package [7,8], according to which the corresponding stresses increase with the increase of strains [6].

The calculations assumed a constant width of the specimens *W* = 25 mm and the spacing of supports equal to *S* = 4·*W* = 100 mm. Using the existing axis of symmetry, it was decided to model only half of the specimen. To reflect the actual behavior of the SEN(B) specimen during experimental studies, the contact problem was solved in the FEM calculations, similarly to the analysis carried out for the stationary fractures [6]. The specimen was loaded by means of a loading pin, modeled as a quarter of an arc with a diameter of *ϕ*10 mm, to which a displacement increasing in time (which was a model loading roller) was applied. The specimen was supported by a support modeled as a half arch, also with a diameter of *ϕ*10 mm, which blocked any displacements. Both contact elements—both the model loading pin and the model of SEN(B) specimen support—respectively a quarter and a half arc, were divided into 90 identical 2-node finite elements (FE) [6].

The numerical model of the SEN(B) specimen used in the calculations was based on the guidelines provided by the authors in [7,8,9,10,11]. The crack tip was modeled as a single node, located exactly in the middle of the specimen width (Figure 3). The near crack tip region was filled with rectangular, nine-node finite elements with nine numerical integration points, 3.57 μm wide and about 0.000267 m high. Since the mesh was filled with nine-node elements, it should be noted that in the initial phase of crack growth, characteristic for small loads, the distance between adjacent nodes in the vicinity of the crack tip was less than 1.8 μm (Figure 3b). In the close vicinity of the crack tip, at a length of 3.75 mm, there were usually 100 identical finite elements—this length guaranteed that the increase in the length of the fracture, in accordance with the *J-R* curves read from experimental studies, would be properly mapped while releasing subsequent nodes with obtaining the appropriate value of the *J*-integral, which is responsible here for controlling the crack length increment. The entire numerical model of the SEN(B) specimen consisted of 9885 nine-node “2D SOLID plane strain” finite elements with mixed interpolation. The total number of nodes in the FEM model was 40,102. The width of the finite element assumed in the vicinity of the crack tip was 7002 times smaller than the width of the specimen *W*, and its height was 94 times smaller.

To represent the full behavior of the material, large deformations and large displacements were assumed in the calculations. The value of the *J*-integral, changing with the external load, was estimated using the “virtual shift method” [7,8], which uses the concept of virtual crack growth to calculate the virtual energy change [7,8]. In the analysis, one integration contour was defined according to the recommendations in [9,10,11]. According to the guidelines provided in [9,10,11], the integration contour was drawn relatively far from the crack tip, preferably through the area dominated by the plane stress state. Such a solution allows to obtain convergent results in the numerically estimated value of the *J*-integral for the case of assuming large deformations and large displacements in the analysis [9,10,11]. Figure 3b shows the dimensions of the integration contour and its position in relation to the crack tip. As can be seen, the contour of integration included all finite elements in a rectangle measuring 12.5 × 15.0 mm. The tests of the influence of the size of the integration contour necessary for the numerical estimation of the *J*-integral, carried out for the purposes of [6], showed that such a size guarantees the convergence of the obtained numerical results, which was mentioned in [6]. This convergence is important because the so-calculated *J*-integral is the “crack driving force”—it is the value of the *J*-integral that decides when the crack length increases in the model of the growing crack, in accordance with the *J*-*R* curve adopted from experimental tests (Figure 1c).

## 3. Analysis of the Obtained Results of Numerical Calculations

After numerical calculations, the following quantities and material behavior were assessed:

(a)Development of the plastic zone with increasing external load.(b)Changes of selected parameters of geometric constraints as a function of the increasing *J*-integral (for selected geometries), defined without the necessity to introduce a constitutive relationship, presented as a function of physical or normalized coordinates at a specific level of external load. These measures include:➢Ratio of effective stress and yield point: $\frac{\sigma_{eff}}{\sigma_{0}}$➢Mean stress and yield point: $\frac{\sigma_{m}}{\sigma_{0}}=\frac{1}{\sigma_{0}}\;\cdot \;\frac{\left(\sigma_{xx}+\sigma_{yy}+\sigma_{zz}\right)}{3}$➢Ratio of mean stress and effective stress: $\frac{\sigma_{m}}{\sigma_{eff}}=\frac{1}{\sigma_{eff}}\;\cdot \;\frac{\left(\sigma_{xx}+\sigma_{yy}+\sigma_{zz}\right)}{3}$➢Stress triaxiality factor, *T_z_*: $T_{z}=\frac{\sigma_{xx}}{\left(\sigma_{yy}+\sigma_{zz}\right)}$(c)Changes in the components of the stress tensor as a function of the increasing *J*-integral (for selected geometries).(d)Changes in the value of the *J*-integral, crack tip opening displacement, external load and the level of maximum stresses opening the crack surfaces as a function of the crack mouth displacement and as a function of the load line displacement.(e)Stress distributions in front of the crack tip, and the above defined geometric constraints measures, plotted as a function of physical coordinates (distance from the crack tip marked by *r*) or as a function of normalized coordinates (normalized distance from the crack tip *ψ* = *r*·*σ*_0_/*J*), for certain external load levels, with these charts drawn up for the moments presented in Table 2.

### 3.1. Analysis of the Development of Plastic Zones during Crack Propagation

Figure 4, Figure 5 and Figure 6 show the distribution of plastic zones in front of the crack tip for three SEN(B) specimens, successively with a thickness of *B* equal to 5, 10 and 15 mm. These graphics were developed for the external loads provided in Table 2. The plastic zone was estimated based on the evaluation of the effective stress values calculated according to the Huber–Misess–Hencky hypothesis at the points of numerical integration of each finite element. Plasticization occurs when the mentioned effective stress is equal to or greater than the yield point.

Figures marked as (a–e) were made with the use of data in the GRAPHER program, while Figures marked as (f) show the plastic zone drawn during postprocessing, using the ADINA environment, for the moment when the value of the *J*-integral was about 75 N mm. This value, as suggested in [5,6], can be treated as a material feature—a measure of the fracture toughness in a plane strain state domination, assuming that its determination is based on the guidelines in [2]. As can be seen, the plastic zone increases with increasing external load. This phenomenon is observed for all specimens considered in the study.

For a specimen with a thickness of *B* = 15 mm, when the *J*-integral reaches the value of approximately 180 N/mm, a significant step increase in the crack length is observed (Figure 1c and Figure 6f). From that moment on, a decrease in the force acting on the specimen is observed, which is accompanied by a decrease in the plastic zone in front of the crack tip (Figure 6e).

The conducted analysis shows that the beginning of the cracking process for 41Cr4 steel, which can be considered the moment when the *J*-integral reaches the value associated with the fracture toughness (in this case 75 N/mm), is accompanied by an earlier, full plasticization of the non-cracked section of the specimen. The fracture process begins when the external load exceeds the limit load by approximately 10%. Such a phenomenon was also observed in the paper of Sumpter and Forbes [12], who analyzed the fracture process of mild steel with a yield point equal to 290 MPa, tested at −50 °C.

### 3.2. Assessment of the Level of Selected Components of the Stress Tensor and Selected Measures of Geometric Constraints

Along with the evaluation of the size and change of plastic zones, for the selected load moments presented in Table 2, for each of the three tested specimens, the level of selected measures of geometric constraints, mentioned above, was assessed. Among them, the following can be distinguished:

Maximum stresses opening the crack surfaces, normalized by the yield point, $\frac{\sigma_{zz}}{\sigma_{0}}$.Ratio of the mean stress and yield point, $\frac{\sigma_{m}}{\sigma_{0}}=\frac{1}{\sigma_{0}}\;\cdot \;\frac{\left(\sigma_{xx}+\sigma_{yy}+\sigma_{zz}\right)}{3}$.Stress triaxiality parameter, defined as the quotient of stresses in the thickness direction and the sum of the other two principal components of the stress tensor, $\frac{\sigma_{xx}}{\left(\sigma_{yy}+\sigma_{zz}\right)}$.Ratio of the effective stress calculated using the HMH hypothesis and yield point, $\frac{\sigma_{eff}}{\sigma_{0}}$.Ratio of the mean stress and effective stress calculated using the HMH hypothesis, $\frac{\sigma_{m}}{\sigma_{eff}}$.

The changes of the above-mentioned quantities were plotted as a function of the normalized distance from the crack tip, calculated as *ψ* = *r*_0_·*σ*_0_/*J*, and as a function of the distance from the initial position of the crack tip, designated as *r*_0_. The results of the analysis are presented in Figure 7, Figure 8, Figure 9, Figure 10 and Figure 11.

Figure 7 shows the changes in the distribution of the maximum stresses opening the crack surfaces, $\frac{\sigma_{zz}}{\sigma_{0}}$. For each of the analyzed thicknesses, it can be noticed that with the increase of the load, accompanied by the increase of the *J*-integral and the increase of the crack length, the maximum stresses opening the crack surfaces slightly increase, reaching an almost steady value. Their value for moments with significant crack growth is *σ_zz_* = 5.0·*σ_0_*. As the fracture length increases, accompanied by an increase in the *J*-integral value, the position of the maximum stresses opening the crack surfaces increases relative to the initial point where the crack tip was located—the maximum stresses opening crack surfaces move away from the initial crack tip position. As can be seen, for each of the thicknesses, a different nature of the change in the distribution of the maximum stresses opening the crack surfaces, $\frac{\sigma_{zz}}{\sigma_{0}}$, for the same material is observed, which is conditioned by completely different *J*-*R* curves that were used to simulate the crack propagation. Summarizing the graphs shown in Figure 7, it can be stated that the thicker the specimen, the further from the crack tip the maximum stresses opening the crack surfaces are in the physical coordinates.

Almost identical conclusions can be drawn by analyzing the distribution of mean stresses normalized by the yield point, $\frac{\sigma_{m}}{\sigma_{0}}$ (Figure 8). With the approach to the crack’s tip, the mean stresses increase, reaching a maximum at a given distance, and then their value decreases—they show the character of changes identical to the maximum stresses opening the crack surfaces. As the external load increases, the value of the maximum mean stresses increases and tends to the value equal to *σ_m_* = 4.0·*σ*_0_. The position of the maximum mean stresses depends on the thickness of the specimen—the thicker the specimen, the further from the initial position of the crack tip the maximum mean stress in the physical coordinates is. The locations of the maximum mean stresses and their values quite significantly depend on the external load.

Figure 9 shows the distributions of the stress triaxiality coefficient defined for the characteristic moments provided in Table 2, defined as $\frac{\sigma_{xx}}{\left(\sigma_{yy}+\sigma_{zz}\right)}$. The graphs indicate that some of the curves are characteristic of the significant crack length increases observed in experimental tests [5,6]. In close proximity to the crack tip, the value of the triaxiality coefficient is equal to 0.5, then as you move away from the fracture tip, the triaxiality coefficient decreases to a value equal to the Poisson’s coefficient, i.e., 0.3. It should be noted that the graphs of changes in the value of the stress triaxiality coefficient have a different character, depending on whether they are plotted as a physical function or a normalized distance from the crack tip. It is left to the reader to interpret this conclusion.

Figure 10 shows the changes of effective stresses calculated according to the HMH hypothesis, with increasing external load and moving away from the crack tip. Near to the crack tip, values of effective stresses at the level of *σ_eff_* = 3.0·*σ*_0_ are observed, and then as you move away from the crack tip, the value of effective stresses drops to the level equal to the yield point. The value of effective stresses equal to or greater than the yield point means that the material in front of the crack is fully plasticized. The decrease in the value of effective stresses below the yield point, visible in Figure 10b,d,f, means that the material has not plasticized yet. The reader can confront these graphs with the appropriate distributions of plastic zones, which are shown in Figure 4, Figure 5 and Figure 6, drawing conclusions individually.

Figure 11 shows the influence of the external load and the distance from the crack tip on the distribution of the quotient of average stresses and effective stresses. The quotient of $\frac{\sigma_{m}}{\sigma_{eff}}$, with the approach to the crack tip, reaches a maximum, the value of which only slightly depends on the thickness of the specimen and the applied external load. There is a slight increase in the maximum value of the quotient $\frac{\sigma_{m}}{\sigma_{eff}}$ with increasing specimen thickness, as well as a quite significant increase with increasing external load. For a specimen with a thickness of *B* = 5 mm, the quotient $\frac{\sigma_{m}}{\sigma_{eff}}$ does not exceed the value equal to 3, while for the thicknesses *B* = 10 and *B* = 15 mm, the quotient $\frac{\sigma_{m}}{\sigma_{eff}}$ for significant increases in the crack length reaches the value equal to 3.5. It can be noticed that the distributions of the quotient $\frac{\sigma_{m}}{\sigma_{eff}}$ in their graphic form resemble the diagrams of changes of average stresses, normalized by the yield point. As the crack crest is approached, the value of the quotient $\frac{\sigma_{m}}{\sigma_{eff}}$ increases, which reaches its maximum, and then the value of this quotient decreases. The position of the maximum of the ratio $\frac{\sigma_{m}}{\sigma_{eff}}$ is closely related to the external load—the higher its value, the further the maximum of the quotient $\frac{\sigma_{m}}{\sigma_{eff}}$ physically is from the initial position of the crack tip.

Figure 12, for one of the geometries considered in the study (specimen with a thickness of *B* = 5 mm), shows how the levels of the principal components of the stress tensor, effective stresses and mean stresses change at different normalized points near the crack tip, depending on the external load expressed by the *J*-integral value. It should be noted that the standardized position in front of the crack tip, marked with *ψ*, was determined in relation to the initial physical location of the crack tip. Along with moving away from the crack tip, a decrease in the level of the main components of the stress tensor and the mean stresses, *σ_m_*/*σ*_0_, is observed. In the very close vicinity of the crack tip for *ψ*∈〈0.5, 1.0〉, an increase in the values of the main components of the stress tensor and mean stresses is observed. Moreover, for *ψ* = 0.5, an almost constant value of effective stresses can be noticed, at the level of *σ_eff_* = 1.5·*σ*_0_, which, at the moment of increasing the crack length by about 0.2 mm, significantly increases to the level of *σ_eff_* = 2.0·*σ*_0_. Within the normalized distances from the initial position of the crack tip, *ψ*∈〈1.0, 3.0〉, effective stresses until the crack length increases by 0.2 mm are practically constant and equal to the yield point. For further normalized distances from the crack tip, the level of effective stresses drops even below the yield point with increasing external load. The reader is left here to draw detailed conclusions from the analysis of Figure 12d, as well as to evaluate the curves shown in Figure 12f, which presents for a specimen with a thickness of *B* = 5 mm changes in the quotient $\frac{\sigma_{m}}{\sigma_{eff}}$ for seven selected normalized distances from the crack tip, with increasing external load, expressed by the *J*-integral value.

The effect of the normalized distance from the initial position of the crack tip and the external load expressed through the *J*-integral on the value of the stress triaxiality coefficient, defined as $\frac{\sigma_{xx}}{\left(\sigma_{yy}+\sigma_{zz}\right)}$, for a specimen with a thickness of *B* = 5 mm, is shown in Figure 13. In close proximity to the crack tip, i.e., *ψ* = 0.5, it can be seen that the level of the stress triaxial coefficient, $\frac{\sigma_{xx}}{\left(\sigma_{yy}+\sigma_{zz}\right)}$, oscillates around the value of 0.5, as for the normalized distance of *ψ* = 1.0, while in the latter case there is a slight decrease in the value of the stress triaxiality coefficient. As you move away from the crack tip, with increasing external load, you can notice a decrease in the level of the stress triaxiality coefficient (almost linear for the normalized distance *ψ* = 2.0), to a value equal to 0.3, which corresponds to the value of the Poisson’s coefficient assumed in the calculations, *ν* = 0.3. It should be noted that the farther it is from the crack tip, the faster the value of the stress triaxiality coefficient, $\frac{\sigma_{xx}}{\left(\sigma_{yy}+\sigma_{zz}\right)}$, reaches the value equal to *ν* = 0.3 (i.e., it occurs at a lower level of the *J*-integral, i.e., at a lower level of external load).

During the numerical calculations, the level of accumulated plastic strains near the crack tip was also assessed for the characteristic moments presented in Table 2 (Figure 14). The level of accumulated plastic strains and their distribution in front of the crack tip depends to a large extent on the specimen thickness and the external load, which for different geometries is accompanied by a different increase in the crack length. The analysis of the distributions presented in Figure 14, combined with the assessment of the developing plastic zones, shows that in the area 1 mm from the crack tip, we are dealing with significant plastic deformations, regardless of the thickness of the specimen. To illustrate the proper level of accumulated plastic strains, in Figure 14h,i, the ordinate has a logarithmic scale. The conclusions drawn from these three additional graphs, based on their analysis and an overview of the developing plastic zones (Figure 4, Figure 5 and Figure 6), are left to the reader.

### 3.3. Analysis of Selected Quantities of Elastic–Plastic Fracture Mechanics

The above analysis of growing plastic zones because of increasing load, along with the assessment of the level of the stress tensor components, accumulated plastic strains or selected measures of geometric constraints, allows us to assess the behavior of 41Cr4 steel during the fracture process. For the three SEN(B) specimens, differing in thickness, the changes in the force, *P,* loading the specimen, the increasing value of the *J*-integral, the increase in the length of the crack, *da*, the increasing crack tip opening displacement, *δ_T_*, and the changing value of the maximum stresses opening crack surfaces, *σ_22_max_*/*σ_0_*, are presented in Figure 15. These graphs were prepared both as a function of the crack mouth displacement, *δ_M_*, which was measured at the edge of the specimen (at the free edge), and as a function of the load line displacement, *v_LL_*, which was dictated by the fact that various normative documents [1,2,3] assume an experimental analysis, based on other recommendations, as to the determination of the energy necessary to estimate the experimental value of the *J*-integral.

In the case of a specimen with a thickness of *B* = 5 mm (Figure 15a,d), initially a stable increase in the crack length can be observed, and then, after reaching the opening of the joint surface at the level of *δ_M_* = 1.5 mm, there is a jump in the length of the crack, which is accompanied by a decreased force loading the specimen, a sharp increase in the *J*-integral and a slight decrease in the value of the maximum stresses opening the crack surfaces. In the remaining analyzed range, the cracking process stabilizes, and the crack length increases quite steadily, which is accompanied by an almost linear increase in the value of the *J*-integral and an increase in the crack tip opening displacement.

A slightly different nature of the changes can be noticed in the case of the analysis of the obtained results for the SEN(B) geometry with a thickness of *B* = 10 mm. While in the case of geometry with a thickness of *B* = 5 mm, a significant step increase in the crack length was observed, amounting to over 1 mm, for geometry with a thickness of *B* = 10 mm, we speak of a step increase of 0.5 mm. This is reflected in Figure 15b,e. During the presented load range, almost linear increases in the crack tip opening displacement, *δ_T_,* and the increase in the fracture length, *da*, both as a function of the crack mouth displacement, *δ_M_*, and the load line displacement, *v_LL_*, are observed. When the displacement of the load line displacement reaches the value *v_LL_* = 1 mm, the crack length increases rapidly—the *J*-integral value increases almost twice. It can be noted that the value of the maximum stresses opening the crack surfaces is almost constant and amounts to approximately (4.5–5.0)·*σ*_0_. This means that in the case of the thickness *B* = 10 mm, the increase in the length of the crack was not so significant as to affect the value of this measure of geometric constraints. It has been shown in [13] that the increase in the crack length is usually accompanied by a decrease in the maximum values of stresses opening the crack surfaces. In the case of the analyzed geometry and the material used, the increase in the fracture length by over 2.5 mm (i.e., by 10% in relation to the width of the specimen, *W*) significantly affects the level of maximum stresses opening the crack surfaces. This conclusion is also confirmed by the evaluation of the cracking process described in the previous paragraph for a specimen with a thickness of *B* = 5 mm (Figure 15a,d).

A stepwise, significant increase in the crack length is observed for a specimen with a thickness of *B* = 15 mm (Figure 15c,f)—it occurs when the load line displacement value, *v_LL_*, is approximately equal to 1.1 mm. It can be noticed then that there is a sharp almost four-fold increase in the value of the *J*-integral, while the value of the force loading the specimen decreases by about 25%. For the entire presented load spectrum, an almost linear increase in the crack length and an increase in the crack tip opening displacement are observed. This specimen is also characterized by an almost constant value of the maximum stresses opening the crack surfaces, amounting to about 4.0·*σ_0_* for the thickness *B* = 15 mm.

Summarizing the briefly discussed course of the fracture process of 41Cr4 steel, it should be noted that the abrupt increases in the crack length are generally accompanied by a rapid increase in the value of the *J*-integral, as well as a decrease in the force loading the specimen. The remaining parameters show rather stable behavior against the rapidly growing crack.

When assessing changes in the basic fracture mechanics parameters, such as the *J*-integral or the crack tip opening displacement, *δ_T_*, the relationship of both quantities with each other through the Shih relationship [14] and the relationship of the *J*-integral with the energy necessary to increase the fracture length cannot be ignored. Shih [14] related the *J*-integral with the crack tip opening displacement, *δ_T_*, with the form relation:(1)$$\delta_{T}=m\;\cdot \;\frac{J}{\sigma_{0}}$$
where *δ_T_* is the crack tip opening displacement, *J* is the *J*-integral, *σ_0_* is the yield point and *m* is the proportionality coefficient, depending on the parameters of the Ramberg–Osgood curve, yield point, Young’s modulus and the stress distribution near the crack tip determined according to the HRR solution—usually it is the value equated with the d_n_ parameter, which is determined by knowing the parameters of the HRR field [15,16] and can also be determined using the computer program presented in [17].

The crack tip opening displacement is determined based on numerical calculations, according to the scheme shown in Figure 16 [18]. The mutual relation of the *J*-integral and the crack tip opening displacement is a linear relationship, where the coefficient of proportionality is the parameter *m*. By dividing Equation (1) on both sides by the length of the non-cracked section of the specimen *b* (where *b* = *W* − *a*), we normalize both sides of Equation (1) [18]:(2)$$\frac{\delta_{T}}{b}=m\;\cdot \;\frac{J}{b\;\cdot \;\sigma_{0}}$$

Denoted by:(3)$$\bar{\delta_{T}}=\frac{\delta_{T}}{b}$$
(4)$$\bar{J}=\frac{J}{b\;\cdot \;\sigma_{0}}$$
the proportionality coefficient, denoted by *m*, linking the crack tip opening displacement, *δ_T_*, and the *J*-integral, can be calculated as:(5)$$m=\frac{\bar{\delta_{T}}}{\bar{J}}$$

As can be seen, the value of the proportionality coefficient, *m,* is equal to the tangent of the angle of inclination of the line $\bar{\delta_{T}}=f\left(\bar{J}\right)$ to the abscissa, which can be written as *m* = *tg*(*φ*) [18]. Figure 17c shows the influence of the specimen geometry on the value of the *m* coefficient. As we can see, the value of this coefficient depends to a very small extent on the thickness of the structural element and varies from the value of 0.55 (*B* = 5 mm) through 0.60 (*B* = 10 mm) to the value of 0.70 (*B* = 15 mm) (Figure 18). Therefore, the conclusions presented in [18] are confirmed, where it has been suggested that the proportionality coefficient, *d_n_*, also denoted as *m*, depends on the geometric constraints that the material opposes the plastic strains developing, both in-plane and out-of-plane constraints, however, a little less. It should be noted that assuming that the material used in the tests—41Cr4 steel—will be described with the appropriate constitutive compounds (Hook’s law and the Ramberg–Osgood relationship), assuming material constants: Young’s modulus *E* = 209 GPa, Poisson’s coefficient *ν* = 0.3, yield point *σ_0_* = 462 MPa, strain hardening constant *α* = 1 and strain hardening exponent *n* = 8.89—and then using the program mentioned in [17], the following parameters of the HRR field are obtained [15,16] for the case of the dominance of the plane strain state, for the direction *θ* = 0: *I_n_* = 4.61, d*_n_* = *m* = 0.477, $\tilde{\sigma_{rr}}$ = 1.72, $\tilde{\sigma_{\theta \theta }}$ = 2.46, $\tilde{\sigma_{eff}}$ = 0.64. It turns out, therefore, that the determined value of the *d_n_* parameter, based on the numerical model and some experimental data, is greater than the value estimated in accordance with the assumptions of the HRR field [15,16].

For many years, the basic formula used to estimate the value of the *J*-integral was in the following form [18,19]:(6)$$J=\frac{\eta \;\cdot \;A}{b\;\cdot \;B}$$
where *b* is the length of non-cracked section of the specimen (*b* = *W* − *a*), *B* is the specimen thickness, *A* is the energy calculated as the area under the force, *P,* curve as a function of the load line displacement, *v_LL_*, or the crack mouth displacement, *δ_M_*, and *η* is a proportionality coefficient, depending on the geometry—for SEN(B) specimens, it usually takes the value of 2. This formula still functions in the standard [2], and many researchers decide to calculate the energy *A* from the graph *P* = *f*(*v_LL_*) or from plot *P* = *f*(*δ_M_*).

By normalizing Equation (6), dividing it on both sides by the product of the length of the non-cracked section of the specimen b and the yield stress σ_0_, the following expression is obtained:(7)$$\frac{J}{b\;\cdot \;\sigma_{0}}=\frac{\eta \;\cdot \;A}{b^{2}\;\cdot \;\sigma_{0}\;\cdot \;B}$$

Denoted by:(8)$$\bar{J}=\frac{J}{b\;\cdot \;\sigma_{0}}$$
(9)$$\bar{A}=\frac{A}{b^{2}\;\cdot \;\sigma_{0}\;\cdot \;B}$$
it can be written that:(10)$$\bar{J}=\eta \;\cdot \;\bar{A}$$
which, after transformation, allows to write the formula for the coefficient *η* [18]:(11)$$\eta =\frac{\bar{J}}{\bar{A}}$$

By plotting the changes of the normalized *J*-integral, denoted by $\bar{J}$, as a function of the normalized energy *A*, denoted by $\bar{A}$ (Figure 17a,b), and then analyzing them, we can confidently say that the value of the coefficient *η* is equal to the tangent of the angel *φ* (the slope of the curve $\bar{J}=f\left(\bar{A}\right)$ to the abscissae) [18]. It can be concluded that the value of the coefficient *η* depends on the thickness of the specimen; therefore, it is confirmed that the geometric constraints, both “in-plane” and “out-of-plane” constraints, affect its value. Figure 18 shows how the value of the coefficient *η* changes as a function of the increasing crack mouth displacement, *δ_M_*, and the increasing load line displacement, *v_LL_*. The increase in load is accompanied by an attempt to obtain a fixed value, and for the thickness *B* = 15 mm, a jump in the value of the parameter *η* is observed, which is related to the step increase in the crack length. The analysis of Figure 18 shows that, for the case where the energy *A* is estimated based on the *P* = *f*(*δ_M_*) diagram (Figure 18a), the value of the coefficient *η* is equal to 3.44, 3.29 and 1.70, for thickness *B* equal to 5, 10 and 15 mm, respectively. Thus, it is observed that the greater the thickness, the smaller the value of the *η* coefficient. The same effect is shown in Figure 18b. If the energy *A* is determined based on the *P* = *f*(*v_LL_*) diagram, the value of the coefficient *η* is equal to 4.72, 4.42 and 2.23, respectively, for the thickness *B* equal to 5, 10 and 15 mm.

Figure 19, Figure 20 and Figure 21, on the example of a specimen with a thickness of *B* = 15 mm, show changes in the force loading the specimen, an increasing value of the *J*-integral, an increase in the crack length, *da*, an increasing crack tip opening displacement and a change in the maximum value of stresses opening the crack surfaces. The first two figures show the changes of the parameters as a function of the crack mouth displacement, *δ_M_*, while the third one presents these changes as a function of the load line displacement, *v_LL_*. In these graphs, a dashed line marks three characteristic moments for which full plasticization of the non-cracked section of the specimen is observed.

The first characteristic moment from the left corresponds to the moment when the value of the *J*-integral is equal to 75 N/mm (which corresponds to the critical value of the *J*-integral determined on the basis of the standard [2]) external load, then slightly, by about 8%, exceeds the permissible limit load for the considered geometry.

The second one corresponds to the hypothetical moment of the crack growth, at which, according to the *J*-*R* curve of the standard [1,2], crack growth begins—the value of the *J*-integral is about 136 N/mm.

The third is the moment when a significant crack growth is observed, amounting to about 3.5 mm, for which the value of the *J*-integral is about 385 N/mm.

## 4. Summary

This paper presented the evaluation of the fracture process of 41Cr4 steel, based on experimental tests and numerical analysis carried out for increasing cracks. Selected results of experimental studies presented in previous publications in 2015–2016 [5], which were used in the construction of FEM models of SEN(B) specimens dominated by a plane strain state, were quoted. Using FEM simulations, the tensile curves and crack growth curves recorded during laboratory tests, the evaluation of plastic zones in the considered specimens, stress distributions, selected measures of geometric constraints, accumulated plastic strains and selected parameters of fracture mechanics were discussed.

Among the most important conclusions resulting from the experimental and numerical analysis of the fracture process of 41Cr4 steel, the following statements can be distinguished:

The beginning of the fracture process for 41Cr4 steel, which can be considered the moment when the *J*-integral reaches the value associated with the fracture toughness (in this case 75 N/mm), was accompanied by an earlier, full plasticization of the non-cracked section of the specimen. The fracture process begins when the external load exceeds the limit load by approximately 10%. Such a phenomenon was also observed in the article of Sumpter and Forbes [12], who analyzed the cracking process of mild steel with a yield point of 290 MPa, tested at −50 °C.

For each of the analyzed thicknesses, it can be noticed that with the increase of the load, accompanied by the increase of the *J*-integral and the increase of the crack length, the maximum stresses opening the crack surfaces slightly increased, reaching an almost steady value. Their value for moments with significant crack growth was *σ_zz_* = 5.0·*σ*_0_. The thicker the specimen, the further from the crack tip the maximum stresses opening the crack surfaces are in physical coordinates.

As the fracture tip was approached, the mean stress increased, reaching its maximum at a given distance, and then its value decreased. As the external load increased, the value of the maximum mean stresses increased and tended to the value equal to σ*_m_* = 4.0·*σ*_0_. The locations of the maximum means stresses and their values quite significantly depend on the external load.

As the crack tip was approached, the value of the ratio $\frac{\sigma_{m}}{\sigma_{eff}}$ increased, which reached its maximum, and then the value of this ratio decreased. The position of the maximum is closely related to the external load—the higher its value, the further the maximum of the ratio $\frac{\sigma_{m}}{\sigma_{eff}}$ physically is from the initial position of the crack tip. The value of the determined ratio, $\frac{\sigma_{m}}{\sigma_{eff}}$, increased with the specimen thickness—from 3.0 to 3.5.

The farther it is from the crack tip, the faster the value of the stress triaxiality coefficient $\frac{\sigma_{xx}}{\left(\sigma_{yy}+\sigma_{zz}\right)}$ reaches the value equal to *ν* = 0.3 (i.e., it occurs at a lower level of the *J*-integral, i.e., at a lower level of external load).

The level of accumulated plastic strains and their distribution in front of the crack tip depends to a large extent on the specimen thickness and the external load, which for different geometries is accompanied by a different increase in the crack length. The analysis of the distributions presented in the paper, combined with the assessment of the developing plastic zones, showed that in the area 1 mm from the crack tip, we are dealing with significant plastic deformations, regardless of the thickness of the specimen.

The jump in increasing the crack was generally accompanied by a sharp increase in the value of the *J*-integral, as well as a decrease in the force loading the specimen. The remaining parameters showed stable behavior against the rapidly growing fracture.

The numerical calculations presented in the paper allow us to estimate the changes of the *J*-integral for the tested steel, the crack tip opening displacement, the size of the plastic zone and the stress distribution in front of the crack tip. Comprehensive numerical analysis, with the use of the actual tensile curve, clearly characterized the behavior of the material under the influence of increasing external loads.

In the future, the author intends to extend the research program with a comprehensive numerical analysis of 3D models, both for stationary and increasing cracks, considering the assumptions of both small and large deformations.

## Figures and Tables

**Figure 1 materials-15-07361-f001:**
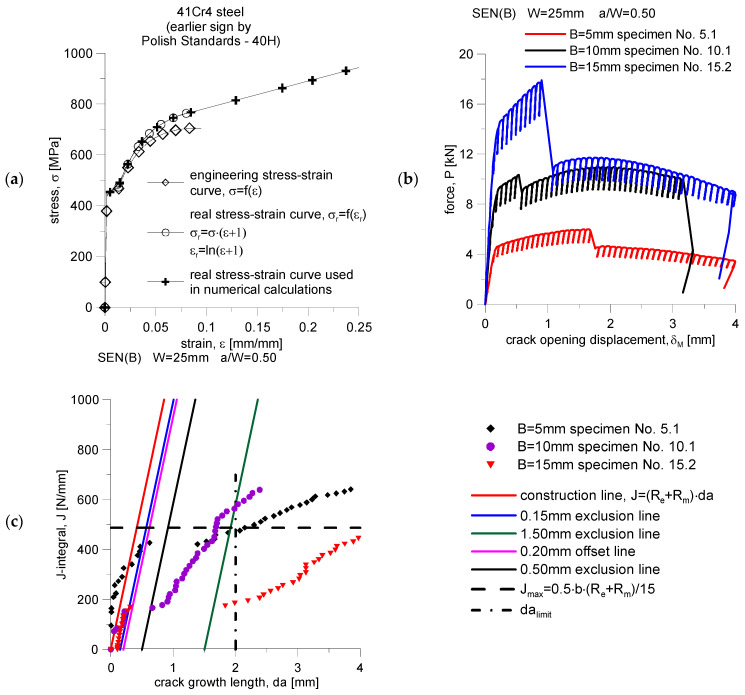
(**a**) The engineering and real tensile curve of 41Cr4 steel, prepared for the needs of numerical FEM calculations [6]. (**b**) Selected diagrams of force, *P,* as a function of crack mouth displacement, *δ_M_*, for steel 41Cr4, recorded using SEN(B) specimens [5,6]. (**c**) The *J*-*R* curves obtained while determining the fracture toughness of 41Cr4 steel—selected results [5,6].

**Figure 2 materials-15-07361-f002:**
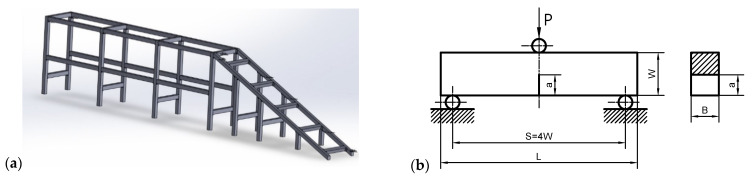
(**a**) Model of the fragment of inroad for cars that has been damaged and for which the research material is presented in [5,6]. (**b**) Geometry of the SEN(B) specimen, which was used in the fracture toughness tests [5,6]: *W*—specimen width (W = 25 mm), *B*—specimen thickness (*B* = {5, 10, 15)} mm), *a*—crack length (*a*/*W* = 0.50), *S—*support spacing, *L*—total length of the specimen (*L* = 120 mm).

**Figure 3 materials-15-07361-f003:**
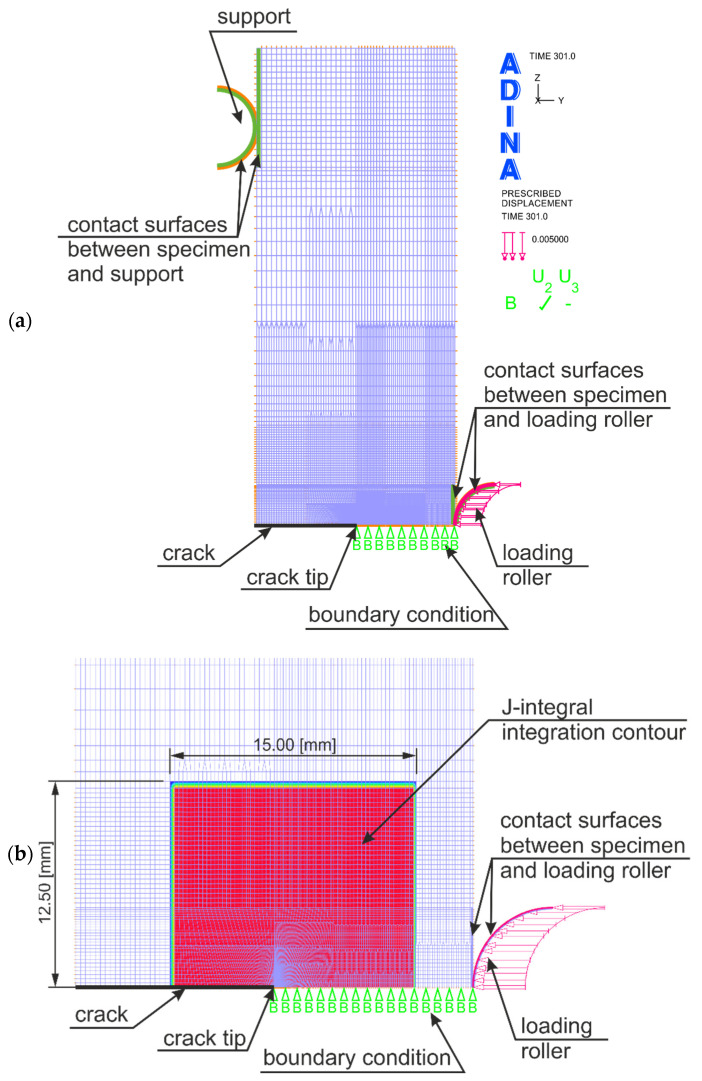
Numerical model of the SEN(B) specimen used in numerical calculations simulating the crack growth: (**a**) model of the whole specimen, and (**b**) enlargement of the area at the top of the crack, along with the designation of the integration contour required for the numerical estimate of the value of the *J*-integral.

**Figure 4 materials-15-07361-f004:**
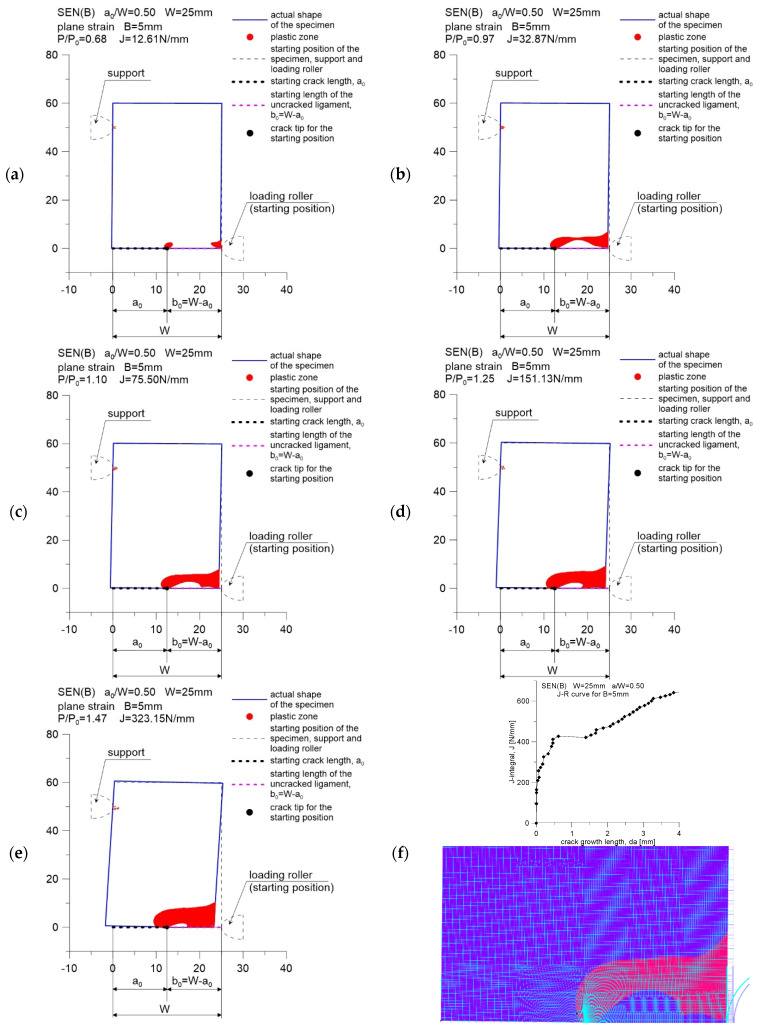
Change in the size of the plastic zone for the SEN(B) specimen, made of 41Cr4 steel, with a thickness of *B* = 5 mm. Specimen number in Table 1: 5.1: (**a**) results for *P*/*P*_0_ = 0.68 and *J*-integral *J* = 12.61 N/m; (**b**) results for *P*/*P*_0_ = 0.97 and *J*-integral *J* = 32.87 N/m; (**c**) results for *P*/*P*_0_ = 1.10 and *J*-integral *J* = 75.50 N/m; (**d**) results for *P*/*P*_0_ = 1.25 and *J*-integral *J* = 151.13 N/m; (**e**) results for *P*/*P*_0_ = 1.47 and *J*-integral *J* = 323.15 N/m; (**f**) the *J*-*R* curve for specimen and the shape of plastic zone for last point from *J-R* curve.

**Figure 5 materials-15-07361-f005:**
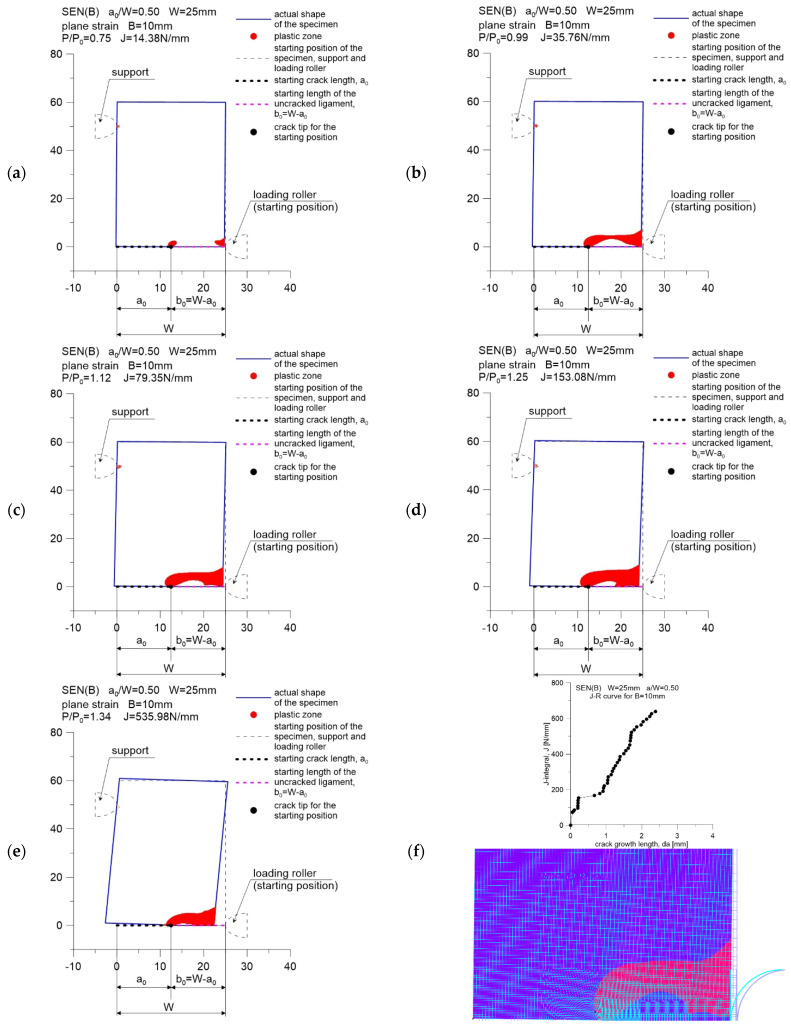
Change in the size of the plastic zone for the SEN(B) specimen, made of 41Cr4 steel, with a thickness of *B* = 10 mm. Specimen number in Table 1: 10.1: (**a**) results for *P*/*P*_0_ = 0.75 and *J*-integral *J* = 14.38 N/m; (**b**) results for *P*/*P*_0_ = 0.99 and *J*-integral *J* = 35.96 N/m; (**c**) results for *P*/*P*_0_ = 1.12 and *J*-integral *J* = 79.39 N/m; (**d**) results for *P*/*P*_0_ = 1.25 and *J*-integral *J* = 153.08 N/m; (**e**) results for *P*/*P*_0_ = 1.34 and *J*-integral *J* = 535.980 N/m; (**f**) the *J*-*R* curve for specimen and the shape of plastic zone for last point from *J-R* curve.

**Figure 6 materials-15-07361-f006:**
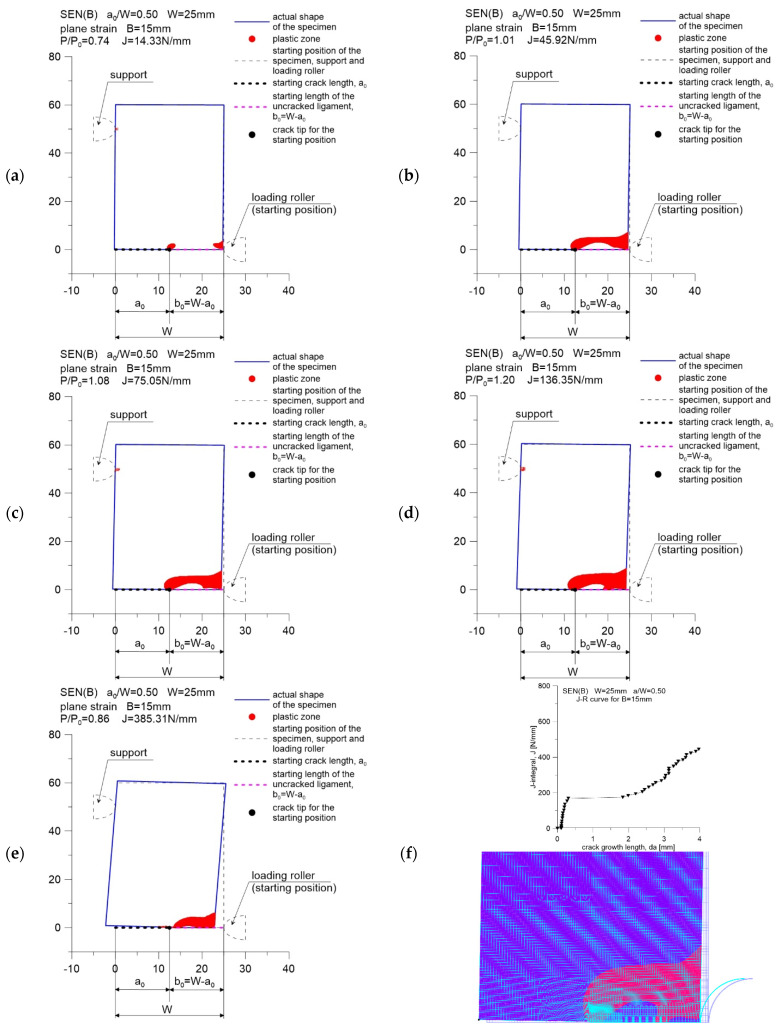
Change in the size of the plastic zone for the SEN(B) specimen, made of 41Cr4 steel, with a thickness of *B* = 15 mm. Specimen number in Table 1: 15.2: (**a**) results for *P*/*P*_0_ = 0.74 and *J*-integral *J* = 14.33 N/m; (**b**) results for *P*/*P*_0_ = 1.01 and *J*-integral *J* = 45.92 N/m; (**c**) results for *P*/*P*_0_ = 1.08 and *J*-integral *J* = 75.05 N/m; (**d**) results for *P*/*P*_0_ = 1.20 and *J*-integral *J* = 136.35 N/m; (**e**) results for *P*/*P*_0_ = 0.86 and *J*-integral *J* = 385.31 N/m; (**f**) the *J*-*R* curve for specimen and the shape of plastic zone for last point from *J-R* curve.

**Figure 7 materials-15-07361-f007:**
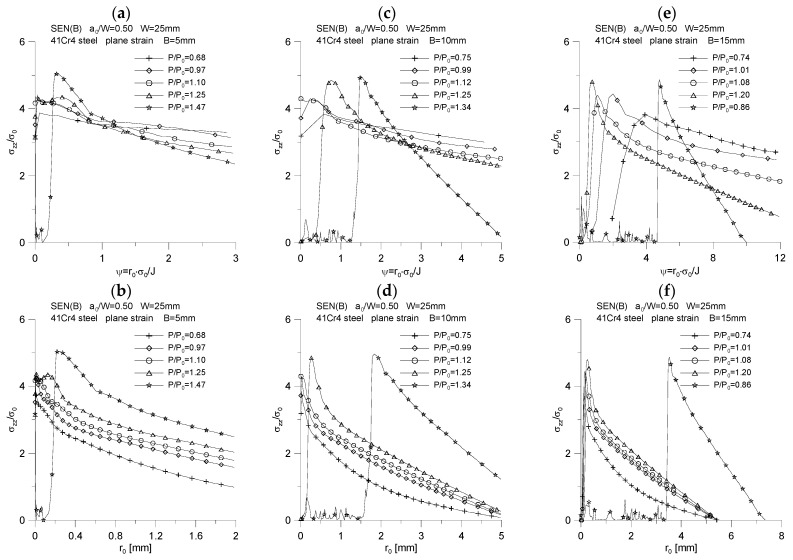
Distribution of maximum stresses opening the crack surfaces, *σ_zz_*/*σ*_0_, for three specimens used in the crack propagation simulation. The graphs (**a**,**c**,**e**) were made for the normalized distance from the crack tip, *ψ* = *r*_0_·*σ*_0_/*J*, while the graphs (**b**,**d**,**f**) were prepared for the physical distance from the initial crack tip position, *r*_0_. The figures show the distributions for the characteristic moments provided in Table 2.

**Figure 8 materials-15-07361-f008:**
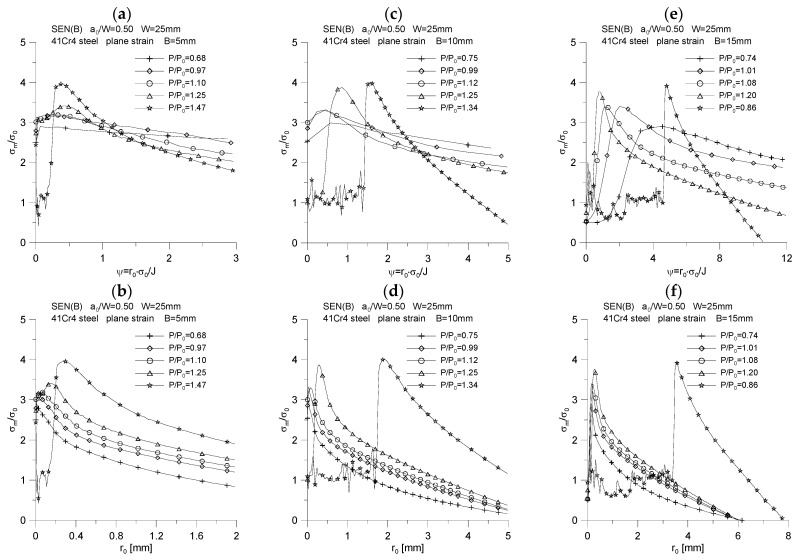
The distribution of mean stresses, *σ_m_*/*σ*_0_, for three specimens used in the crack propagation simulation. The graphs (**a**,**c**,**e**) were made for the normalized distance from the crack tip, *ψ* = *r*_0_·*σ*_0_/*J*, while the graphs (**b**,**d**,**f**) were prepared for the physical distance from the initial crack tip position, *r_0_*. The figures show the distributions for the characteristic moments provided in Table 2.

**Figure 9 materials-15-07361-f009:**
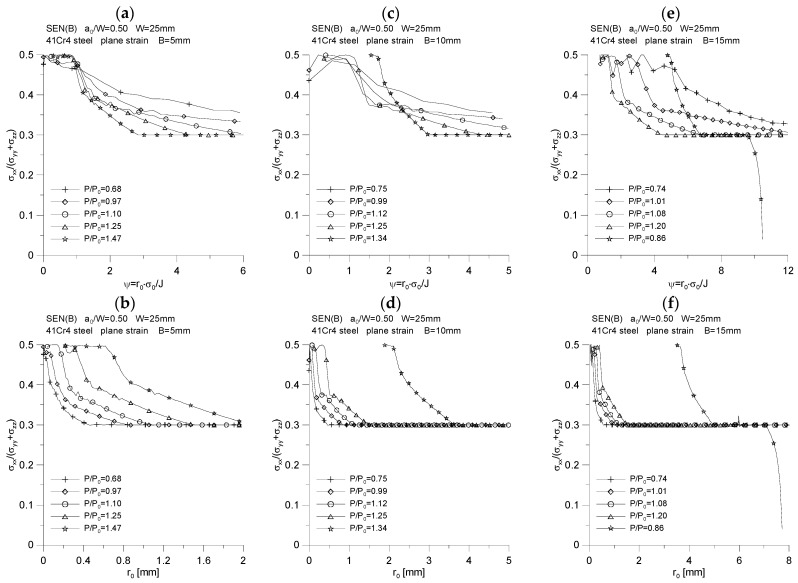
Distribution of the stress triaxiality coefficient, *σ_xx_*/(*σ_zz_* + *σ_yy_*), for three specimens used in the crack propagation simulation. The graphs (**a**,**c**,**e**) were made for the normalized distance from the crack tip, *ψ* = *r*_0_·*σ*_0_/*J*, while the graphs (**b**,**d**,**f**) were prepared for the physical distance from the initial crack tip position, *r*_0_. The figures show the distributions for the characteristic moments provided in Table 2.

**Figure 10 materials-15-07361-f010:**
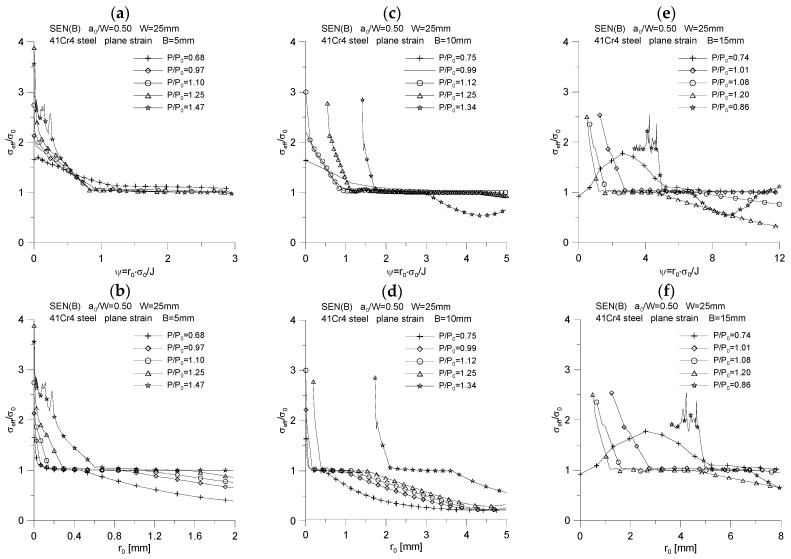
The distribution of effective stresses, *σ_eff_*/*σ*_0_, for three specimens used in the crack propagation simulation. The graphs (**a**,**c**,**e**) were made for the normalized distance from the crack tip, *ψ* = *r*_0_·*σ*_0_/*J*, while the graphs (**b**,**d**,**f**) were prepared for the physical distance from the initial crack tip position, *r*_0_. The figures show the distributions for the characteristic moments provided in Table 2.

**Figure 11 materials-15-07361-f011:**
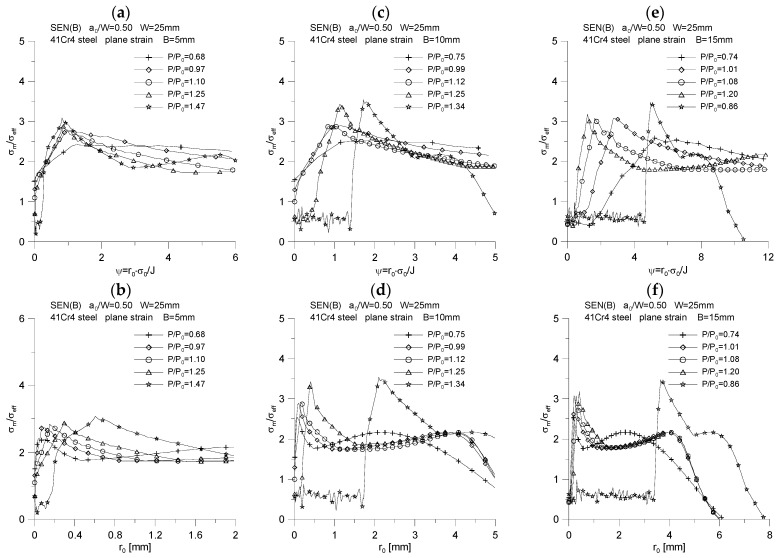
Distribution of the quotient *σ_m_*/*σ_eff_* for three specimens used in the crack propagation simulation. The graphs (**a**,**c**,**e**) were made for the normalized distance from the crack tip, *ψ* = *r*_0_·*σ*_0_/*J*, while the graphs (**b**,**d**,**f**) were prepared for the physical distance from the initial crack tip position, *r*_0_. The figures show the distributions for the characteristic moments provided in Table 2.

**Figure 12 materials-15-07361-f012:**
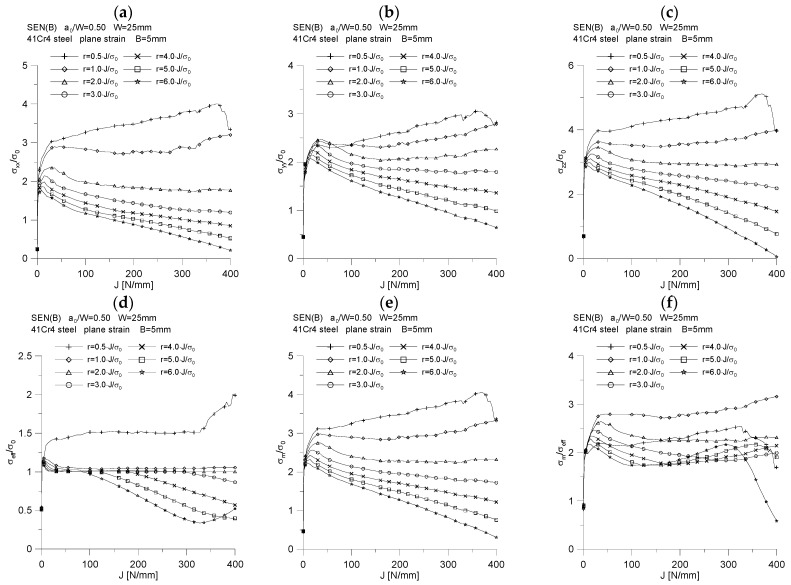
Influence of the external load and the normalized distance from the initial position of the crack tip on the distribution of the main components of the stress tensor (**a**,**b**,**c**), the distribution of effective stresses (**d**), mean stresses (**e**) and the ratio *σ_m_*/*σ_eff_* (**f**), for a SEN(B) specimen with a thickness of *B* = 5 mm, made of 41Cr4 steel.

**Figure 13 materials-15-07361-f013:**
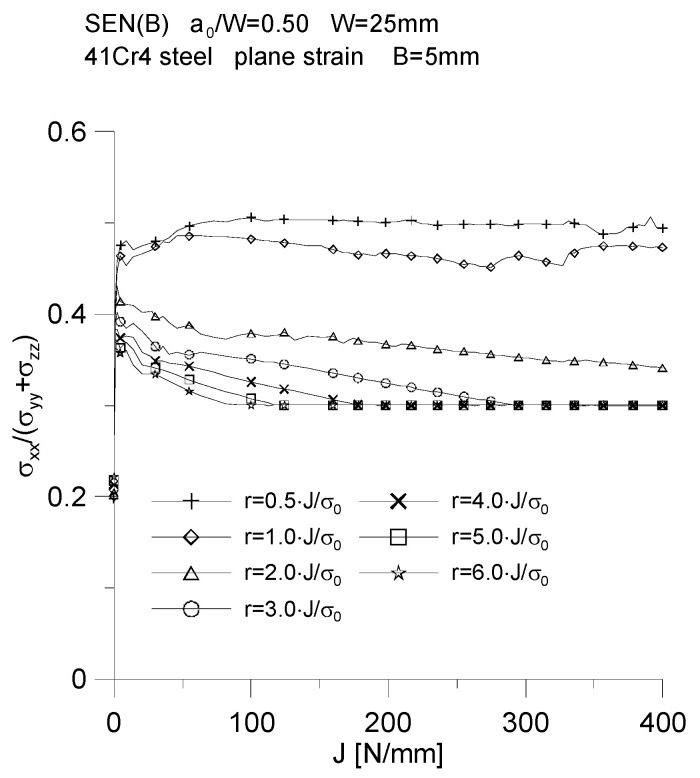
Influence of the external load and the normalized distance from the initial position of the crack tip on the distribution of the stress triaxiality coefficient, *σ_xx_*/(*σ_zz_* + *σ_yy_*), for the SEN(B) specimen with a thickness of *B* = 5 mm, made of 41Cr4 steel.

**Figure 14 materials-15-07361-f014:**
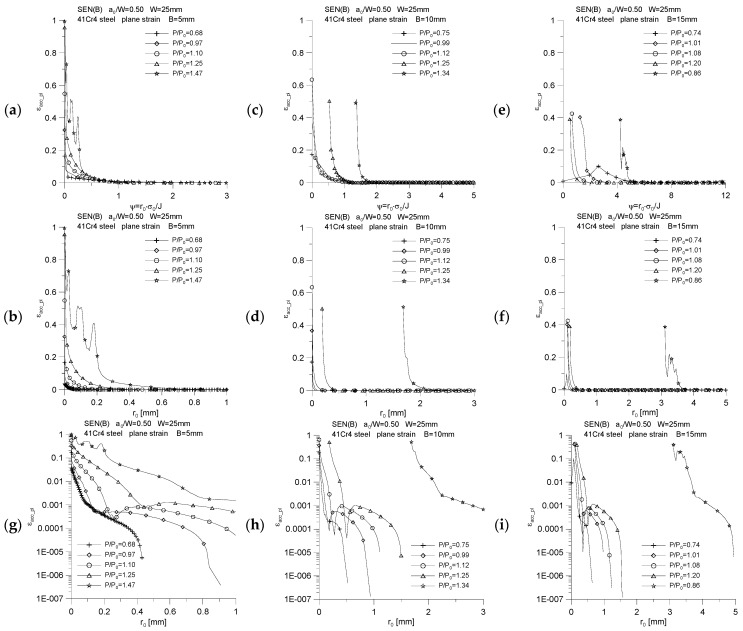
Distribution of accumulated plastic strains, *ε_acc_pl_*, near the crack tip, for three specimens used in the crack propagation simulation. The graphs (**a**,**c**,**e**) were made for the normalized distance from the crack tip, *ψ* = *r*_0_·*σ*_0_/*J*, while the graphs (**b**,**d**,**f**) were prepared for the physical distance from the initial crack tip position, *r*_0_. In the graphs marked with (**g**,**h**,**i**), the ordinate scale is presented in logarithmic form to illustrate the level of accumulated plastic strains more accurately in front of the crack tip, for different levels of external load. The figures show the distributions for the characteristic moments provided in Table 2.

**Figure 15 materials-15-07361-f015:**
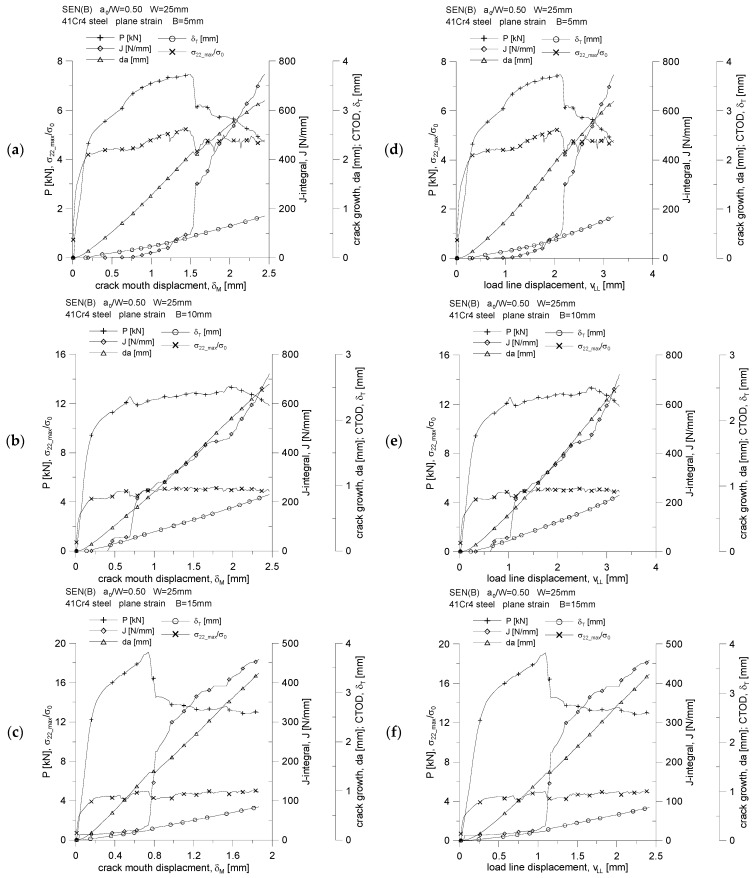
Changes of: the force, *P,* loading the specimen, the value of the *J*-integral, the crack length increment, *da*, the crack tip opening displacement, *δ_T_*, and the maximum stresses opening the crack surfaces, *σ_22_max_*/*σ*_0_, as a function of the crack mouth displacement, *δ_M_* (**a**–**c**), and as a function of load line displacement, *v_LL_* (**d**–**f**).

**Figure 16 materials-15-07361-f016:**
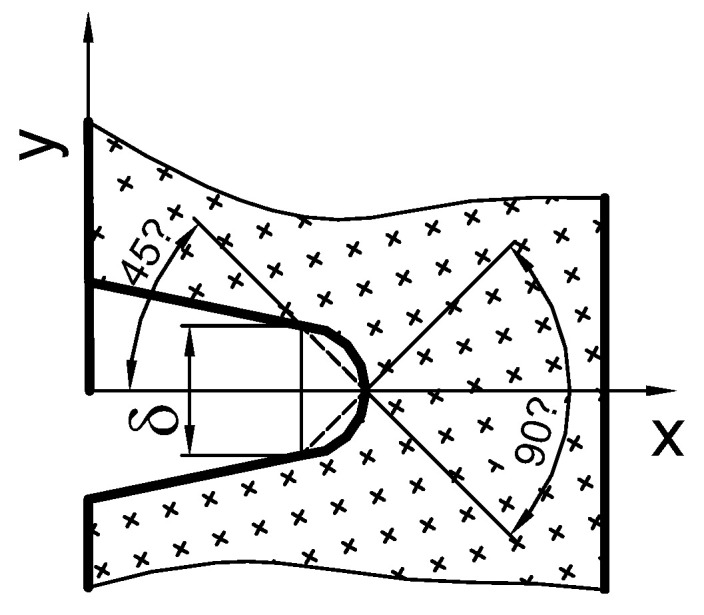
Shih’s concept [14] used to determine the crack tip opening displacement [18].

**Figure 17 materials-15-07361-f017:**
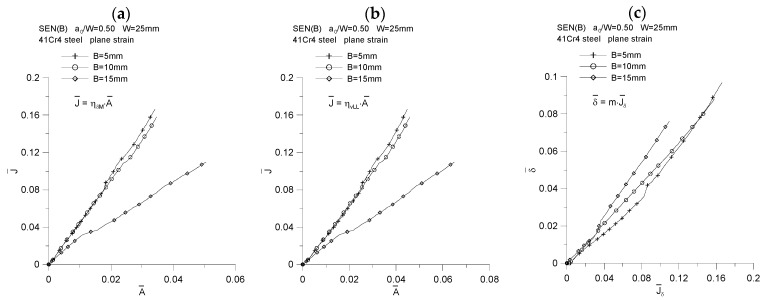
The influence of the specimen thickness on the value of the proportionality coefficient between the *J*-integral and the energy required to increase the crack length (**a**,**b**) and on the proportionality coefficient between the crack tip opening displacement and the *J*-integral (**c**), for the SEN(B) specimen made of 41Cr4 steel.

**Figure 18 materials-15-07361-f018:**
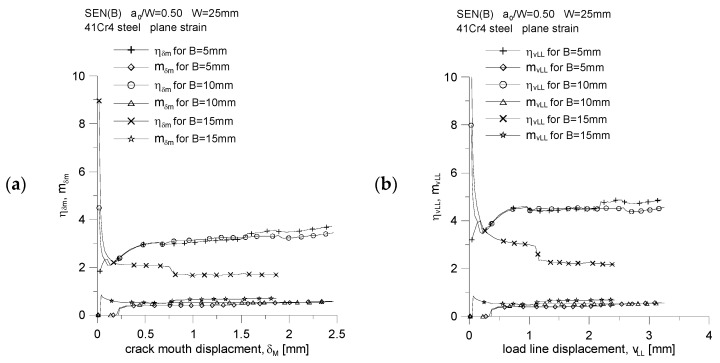
Influence of the specimen thickness on the value of the proportionality coefficient between the *J*-integral and the energy required to increase the crack length and on the proportionality coefficient between the crack tip opening displacement and the *J*-integral, for SEN(B) specimens made of 41Cr4 steel: (**a**) the energy values, *A,* were determined based on the plot *P* = f(*δ_M_*), and (**b**) the values of energy *A* were determined on the basis of the diagram *P* = *f*(*v_LL_*).

**Figure 19 materials-15-07361-f019:**
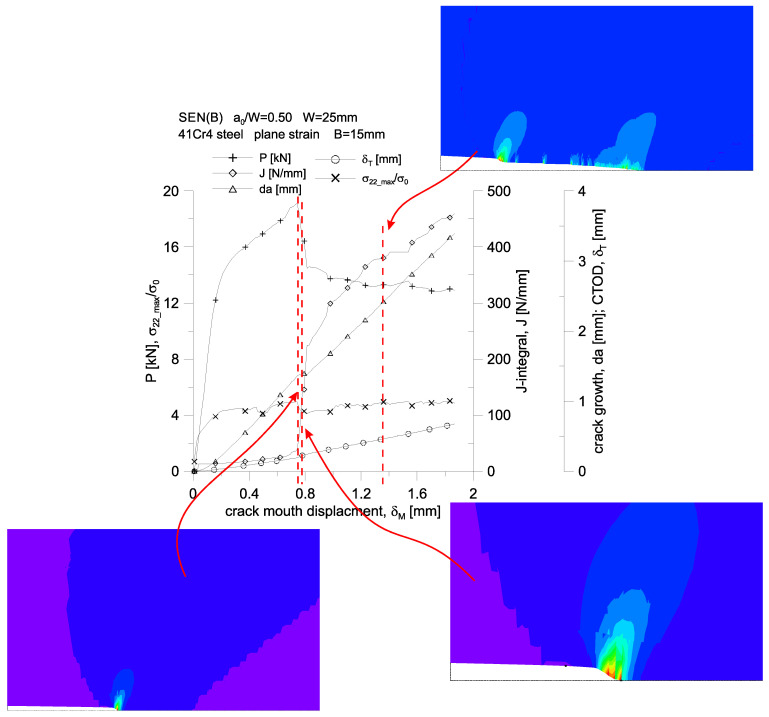
Changes of: the force, *P,* loading the specimen, the value of the *J*-integral, the crack length increment, *da,* the crack tip opening displacement, *δ_T_*, and the maximum stresses opening the crack surfaces, *σ_22_max_*/*σ*_0_, as a function of the crack mouth displacement, *δ_M_*, together with the distribution of accumulated plastic strains in close proximity to the crack tip. 41Cr4 steel specimen with a thickness of *B* = 15 mm.

**Figure 20 materials-15-07361-f020:**
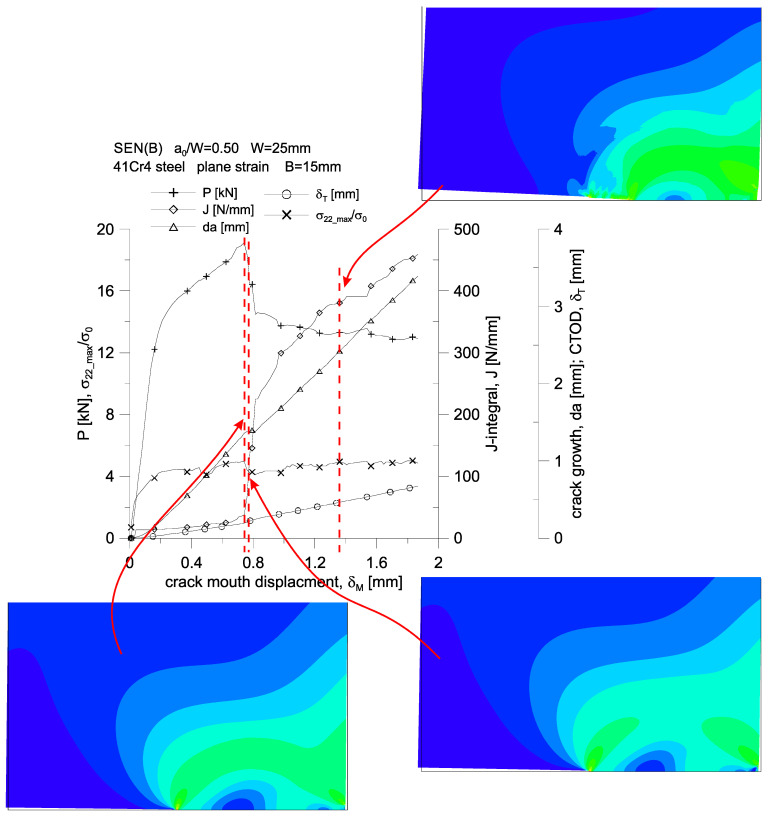
Changes of: the force, *P,* loading the specimen, the value of the *J*-integral, the crack length increment, *da*, the crack tip opening displacement, *δ_T_*, and the maximum stresses opening the crack surfaces, *σ_22_max_*/*σ*_0_, as a function of the crack mouth displacement, *δ_M_*, together with the distribution of effective stresses in close proximity to the crack tip. 41Cr4 steel specimen with a thickness of *B* = 15 mm.

**Figure 21 materials-15-07361-f021:**
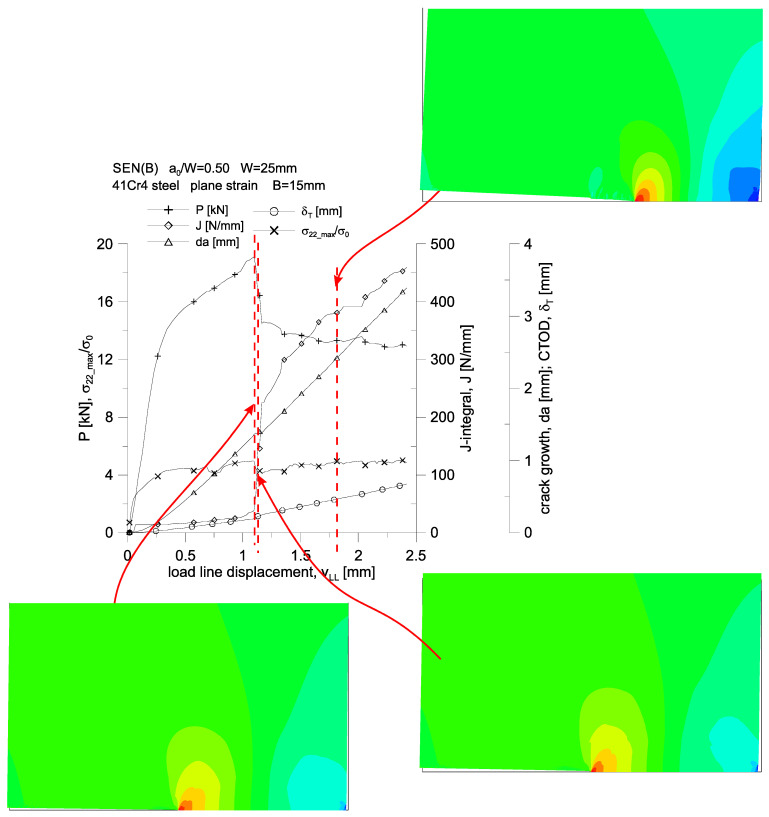
Changes of: the force, *P,* loading the specimen, the value of the *J*-integral, the crack length increment, *da*, the crack tip opening displacement, *δ_T_*, and the maximum stresses opening the crack surfaces, *σ_22_max_*/*σ*_0_, as a function of the load line displacement, *v_LL_*, together with the distribution of crack opening stresses, in close proximity to the crack tip. 41Cr4 steel specimen with a thickness of *B* = 15 mm.

**Table 1 materials-15-07361-t001:** Summary of selected uniaxial tensile test results and J_C_ fracture toughness tests for 41Cr4 steel based on [5,6].

41Cr4 Steel	*E* (GPa)	*R_eh_* (MPa)	*R_m_* (MPa)	*A_t_*	*ε_m_*	*n*
average	209	462	707	0.142	0.096	8.89
minimum	202	450	703	0.130	0.088	8.28
maximum	216	473	714	0.153	0.105	9.66
median	209	460	705	0.144	0.098	8.90
No.	*B* [mm]	*P_KQ_* [kN]	*J_KQ_* [N/mm]	*P_Q_*/*P*_0_	PN-88/H-04336	
					*J_Q_* [N/mm]	25⋅*J_Q_*/*σ*_0_
5.1	5	3.30	6.56	0.68	328.71	17.79
10.1	10	8.00	9.59	0.81	45.74	2.48
15.2	15	11.10	8.06	0.74	74.30	4.02

**Table 2 materials-15-07361-t002:** Values of the *J*-integral, increments of crack length, for selected moments in the range of the entire load of SEN(B) specimens for 41Cr4 steel.

*B* = 5 mm			*B* = 10 mm			*B* = 15 mm		
*P*/*P*_0_	*da* (mm)	*J* (N/mm)	*P*/*P*_0_	*da* (mm)	*J* (N/mm)	*P*/*P*_0_	*da* (mm)	*J* (N/mm)
0.68	0	12.60	0.75	0	14.38	0.74	0	14.33
0.97	0	32.87	0.99	0	35.76	1.01	0	45.92
1.10	0	75.50	1.12	0	79.35	1.08	0	75.05
1.25	0.007	151.13	1.25	0.210	153.08	1.20	0.200	136.25
1.47	0.200	323.15	1.34	1.770	535.98	0.86	3.480	385.31

## Data Availability

Not applicable.

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
