# Peer review of "Numerical Evaluation of the Fracture Process of 41Cr4 Steel: Analysis of Cracks Grown in a Plane Strain State Domination Based on Experimental Results"

_materials, 2022, doi:10.3390/ma15207361_

Round 1
Author Response
REVIEW No. 1:
In accordance with the comments of the Honorable Reviewers, the article was sent to a linguistic proofreading
REVIEW No.1 - note 1: The presented work emphasized on the investigation on fracture behaviour of steel alloy through experimental and numerical approaches. However, following shortcomings and serious concerns are presented in the manuscript and which has to be rectified before further processing. I request mandatory revision, as listed below, please do not simply respond but revise manuscript.
RESPONSE: Appropriate corrections were added to the revised version of the paper. The research that had been carried out before was mentioned, reference was made to it, and a commentary on the genesis of the paper was added - where did the idea for researching the material come from. The section about the using the 41Cr4 steel in industry was added. New figure was added to manuscript, which presets the Everything was corrected as suggested by the Honorable Reviewer.
REVIEW No.1 - note 2:The similarity is found to be 35%. Most of the sentences are copied from existing studies. Therefore, it is necessary to reduce the similarity to below 20%.
RESPONSE: The manuscript is an author's scientific description. If the Dear Reviewer believes that it is copied from other papers, please indicate the source and the tool to be checked. I will check it. Initially, the manuscript was written in Polish, then it was translated into English individually by the author. The version delivered for re‑review is the version after linguistic proofreading performed by a professional translation agency. If there are any similar elements, they come from two of my own publications (also my own), which I refer to in the text and I mention them even in the first chapter, which was developed on these two articles. The remaining part - the description of the numerical model, the description of the results and the analysis (I also refer to my paper here) was first prepared in Polish as the mother tongue, and then I translated it into English.
REVIEW No.1 - note 3:Since, numerous existing works were available in the same filed of fracture investigation through numerical approach. Authors should explicitly state the novel contribution of this proposed work and also enumerate the practical benefit of the presented work for industries and researchers.
RESPONSE: It was noted in the text that it is a supplement to the previous two publications focused only on experimental research and on the assessment of the material's behavior during fracture, assuming that the stationary crack is considered. The approach to the assessment of the moment of fracture is innovative - not entirely in line with the rather conservative standards given in the subject standards. Manuscript for industry can serve as an aid in solving similar problems. You don't always know all the characteristics of the material. And when you evaluate something, something that has been destroyed, sometimes you need to have a recipe for doing it - this paper shows how comprehensive the analysis should be. This is indicated in the text.
REVIEW No.1 - note 4:The abstract should provide an overview of proposed methods/methodology, materials with obtained results in the form of quantitative values. The present abstract is looks like Introduction. Therefore, it is mandatory to include the quantitative analysis results of proposed work in Abstract section.
RESPONSE: The abstract may look like an introduction, but in general it is acceptable. I review a lot myself, meet many forms of abstracts. In my opinion, it should indicate what will be at manuscript. In this article, there are many quantitative and qualitative results - therefore they are not quoted here, but only an indication of what will be part of the paper.
REVIEW No.1 - note 5:For readers to quickly catch the contribution in this work, it would be better to highlight major difficulties and challenges, and authors' original achievements to overcome them, in a clearer way in Introduction section.
RESPONSE: Yes, Dear Review is right here. I was taught by my scientific mentors not to boast about my achievements, which is why this form of introduction was adopted. In my humble opinion, the reader should judge the author's contribution and commitment for himself.
REVIEW No.1 - note 6:The results and discussion are not clearly dealt the outcomes of the proposed work. The authors should explicitly state the novel contribution of this work, the similarities, and the differences of this work with the previous publications in this section.
RESPONSE: The ¾ of the manuscript volume was devoted to the presentation of the obtained numerical results and their analysis. It is 16 pages of text with 18 Figures. So I do not understand this remark. Nobody has studied such material in such a way, and it is not possible to refer to anyone's results.
REVIEW No.1 - note 7:It is suggested to highlight the limitations of this study, suggested improvements of this work and future directions in the conclusion section. Also, the conclusion can be presented better than the present form with more findings. The Conclusion (Summery) is not exactly correct as it is like Discussions. It should be modified and represents the exact outcomes of the work.
RESPONSE: There have been added provisions regarding other foreseen activities for the assessment of the material under consideration. The author intends to carry out a comprehensive program of 3D numerical calculations, where he will estimate many parameters in the field of fracture mechanics, both on the assumptions small and large deformations.
REVIEW No.1 - note 8:Reference is not in the exact formatting of “Materials”. It should be modified accordingly. Please note that the comments are intended merely to assist the authors in improving the manuscript and ensuring that published papers are of the highest quality. They are in NO WAY intended to discourage or demean the authors personally.
RESPONSE: Thank you to the Honorable Reviewer for such constructive comments. Certainly the manuscript will be much better. I tried to improve the paper in accordance with the guidelines of three independent Reviewers. Sometimes it's hard to include all the corrections - I did it whenever possible. The article was not prepared on the MDPI Materials Journal template. The version sent to presenters is automatically processed by the system - it seems to me. In case of positive opinions and approval of the paper for publication, the entire text will be formatted according to the guidelines of the MDPI Materials Journal. Literature references will also be properly formatted in accordance with the Publisher's guidelines.
I assume that my comments and explanations have dispelled all doubts of the Honorable Reviewer. I believe that the corrections have made the manuscript more valuable and can be recommended for publication.
Reviewer 2 Report
The problem under consideration was investigated by many authors about fifty years ago, and in the manuscript there are no advances on the problem.
I am very doubtful about the chosen numerical strategy. The FEM model is not clearly described, but the following aspects are the most critical:
1) 3D effects cannot properly represented with 2D models
2) Applied symmetry boundary conditions do not allow for crack propagation, therefore it is not possible to understand what is actually calculated. As a consequence, the plastic zone in front of the crack tip is completely unrealistic (elastic region right in front of the crack tip). There is no clearly defined crack propagation criterium.
3) The model complexity is unnecessarily increased by adding over detailed modeling of contact in correspondence of supports, large deformation etc.
The manuscript, in the present form, is far from suitable for publication
Author Response
REVIEW No. 2:
In accordance with the comments of the Honorable Reviewers, the article was sent to a linguistic proofreading performed by a professional translation agency.
REVIEW No.2 - note 1:The problem under consideration was investigated by many authors about fifty years ago, and in the manuscript there are no advances on the problem.
RESPONSE: Indeed, the problem of material behavior during the fracture process has been analyzed from the beginning of fracture mechanics - Williams' solution given in 1957. Manuscript will not add anything new to the research procedures themselves, nor will it indicate new research methods. Manuscript aims to present a numerical assessment of the physical behavior of the material obtained from a structure that has been damaged, indicating certain paths used by many researchers that sometimes allow us to break out of a situation in which we cannot, according to generally accepted standards, assess the fracture toughness or strength of the structure containing the defect. This paper is also aimed at people who may have encountered a similar problem. The series of numerical analyzes presented in the paper indicates that the evaluation of the fracture process is not only laboratory work, but also a numerical analysis that allows to assess selected parameters of fracture mechanics, measurements of geometric constraints or the level of stress tensor components. Moreover, based on the performed numerical analyzes, attempts were made to estimate the value of the proportionality coefficient between the J integral and the crack tip opening in the Shih formula and the proportionality coefficient between the J-integral and the energy necessary to increase the crack length according to Rice's formula. The paper illustrates the procedure and gives the values of these coefficients for a specific, tested material.
REVIEW No.2 - note 2:I am very doubtful about the chosen numerical strategy. The FEM model is not clearly described, but the following aspects are the most critical:
RESPONSE: The description of the numerical model is presented on 1.5 typewritten pages plus a Figure for the entire page. The numerical model has been described in detail, the applied boundary conditions and the load to the specimen have been shown and explained, the structure of the finite element mesh has been discussed, the sizes of finite elements in the vicinity of the crack tip have been given, as well as the method of numerical determination of the J-integral - the defined integration contour has been discussed. The numerical model was created according to the recommendations of prof. Brocks given in literature [9, 10], commonly accepted. In the author's humble opinion, the presented numerical model is more thoroughly discussed than the presentation of similar elements in many publications issued by MDPI.
REVIEW No.2 - note 3: 1) 3D effects cannot properly represented with 2D models
RESPONSE: I agree with the opinion of the Honorable Reviewer that 3D effects in 2D - plane problems cannot be fully presented properly. However, in the analyzed analysis, we deal with the dominance of a plane strain state, the component of the stress tensor in the direction of the third axis - here, in the thickness direction is non-zero. Therefore, it is worth considering such parameters as the stress triaxial coefficient , effective stresses , mean stresses effective stresses and mean stresses . These values have been considered in the scientific literature many times by many researchers - Sumpter, O'Dowd, Shih, Henry etc. Moreover, these parameters can be successfully used in the proposal of various fracture criteria, as demonstrated by e.g. Guo Wanlin. As is well known, the fracture toughness is determined in the plane strain state domination, and the analysis of the quantities listed in the paper is justified. So I consider the objection to be necessary for discussion, however it should not disqualify my manuscript.
REVIEW No.2 - note 4: 2) Applied symmetry boundary conditions do not allow for crack propagation, therefore it is not possible to understand what is actually calculated. As a consequence, the plastic zone in front of the crack tip is completely unrealistic (elastic region right in front of the crack tip). There is no clearly defined crack propagation criterium.
RESPONSE: I do not agree with the opinion of the Honorable Reviewer. The boundary conditions are applied in accordance with the applicable rules, in accordance with the instructions of the ADINA System, they allow the crack length to increase with the increase of the J-integral. which is the crack growth control parameter. The J-R curve was derived from the experimental studies described in the paper. It was clearly stated in the paper that the crack propagation criterion is based on the J-R curve recorded during the conducted experimental tests. I also disagree with the opinion that it is not known what counts at the moment - everything is planned, the model is physically justified, it is known what is calculated in the model. In each model calculated in the ADINA System, we then have at our disposal a failure zone in the results, components of the displacement vector, components of strain and stress tensors, effective stresses, accumulated plastic strains, J-integral, etc. The paper clearly states what is calculated. The comment from the Honorable Reviewer is out of place in my opinion. The crack propagation criterion is clearly indicated. The shape of the plastic zone in the case of the dominance of the plane strain state is different than for the dominance of the plane stress state or for the 3D cases. The answer to this can be found in Anderson's „Fracture Mechanics” handbook. Moreover, the plastic zone will be different for numerical considerations conducted at a stationary fracture and different for cases of increasing crakcs. The plastic zone generally grows with the load, but when the crack propagates, its length changes and it is noted that the plastic zone may change its size - the crack will increase, so the material is weaker and the external load decreases, which was also observed during the described laboratory tests . It therefore seems that the results are correctly presented and interpreted correctly.
REVIEW No.2 - note 5: 3) The model complexity is unnecessarily increased by adding over detailed modeling of contact in correspondence of supports, large deformation etc.
RESPONSE: Simplifying the support or load roller would affect the results obtained in the calculation solver. The author's experience shows that the adopted calculation model is correct and the calculation method is recommended by the authors of the ADINA system. In addition, the presented calculation scheme is the result of many years of work in the ADINA system, guarantees the convergence of results, and is also acceptable to other Authors and Reviewers of scientific papers. With the current high computing power of computers, fast processors, disks and large operating memory, the computation time is not extended if the simplifications mentioned by the Dear Reviewer are not applied. I believe that contact modeling in solving these issues is very important and reflects the physical behavior of the sample, which is observed during experimental research in the laboratory. As I have already mentioned, the method of modeling the support and load in the case of contact issues - and this is what we deal with in research with the use of SEN (B) samples, is recommended by the creators of the ADINA system, and is also practiced by many researchers dealing with similar issues.
I assume that my comments and explanations have dispelled all doubts of the Honorable Reviewer. I believe that the corrections have made the manuscript more valuable and can be recommended for publication.

Reviewer 3 Report
In this paper, the author evaluated the fracture process of 41Cr4 steel, based on experimental tests and numerical analysis carried out for increasing cracks. Using FEM simulations, the tensile curves and crack growth curves recorded during laboratory tests, the evaluation of plastic zones in the considered specimens, stress distributions, selected measures of geometric constraints, accumulated plastic strains and selected parameters of fracture mechanics, was discussed.
The paper is interesting, well written and well structured. It deals with an important problem related to the numerical analysis of the crack growth of 41Cr4steel, based on a properly constructed numerical model, considering the actual experimental tensile curve, as well as the actual J-R curve - crack growth curve, presenting changes in the value of the J-integral as a function of crack length increments da, which can be written as J=f(da). The text is clear and easy to read, and the results are sufficiently discussed. Overall, the manuscript is well thought out and written, the objectives clearly stated, simulation and experimental methods are advanced, data statistically analyzed, the conclusions well supported by the data presented. This makes the paper possible for publishing in the Materials. However, to accept, the paper should be subject to some minor revision. The detailed remarks are as follows:
1. In my opinion the word “Experimental” in Title should be removed. Since, in this paper the author just performed numerical evaluation, which was compared with the experimental results performed in the previous papers.
2. In Abstract and Summary sections, please remove the citation.
3. In introduction section the author should provide more research that have been done in this field to provide sufficient background and relevant references. Sufficient information about the previous study findings should be added and compared with the present study so that the readers could follow the present study rationale and procedures.
4. Please explain more about the material 41Cr4steel. Why did the author choose this material and where this material is usually applied in the industry?
Author Response
REVIEW No. 3:
In accordance with the comments of the Honorable Reviewers, the article was sent to a linguistic proofreading performed by a professional translation agency.
REVIEW No. 3 - note 1: In this paper, the author evaluated the fracture process of 41Cr4 steel, based on experimental tests and numerical analysis carried out for increasing cracks. Using FEM simulations, the tensile curves and crack growth curves recorded during laboratory tests, the evaluation of plastic zones in the considered specimens, stress distributions, selected measures of geometric constraints, accumulated plastic strains and selected parameters of fracture mechanics, was discussed.
RESPONSE: Dear Reviewer, thank You for the positive opinion of my paper and for the time devoted to its analysis. I appreciate it.
REVIEW No. 3 - note 2:The paper is interesting, well written and well structured. It deals with an important problem related to the numerical analysis of the crack growth of 41Cr4steel, based on a properly constructed numerical model, considering the actual experimental tensile curve, as well as the actual J-R curve - crack growth curve, presenting changes in the value of the J-integral as a function of crack length increments da, which can be written as J=f(da). The text is clear and easy to read, and the results are sufficiently discussed. Overall, the manuscript is well thought out and written, the objectives clearly stated, simulation and experimental methods are advanced, data statistically analyzed, the conclusions well supported by the data presented. This makes the paper possible for publishing in the Materials. However, to accept, the paper should be subject to some minor revision. The detailed remarks are as follows:
RESPONSE: Dear Reviewer, thank You for the positive opinion of my paper and for the time devoted to its analysis. I appreciate it.
REVIEW No. 3 - note 3: 1. In my opinion the word “Experimental” in Title should be removed. Since, in this paper the author just performed numerical evaluation, which was compared with the experimental results performed in the previous papers.
RESPONSE: The title of the work has been changed. The new manusript title is as follows: „Numerical evaluation of the fracture process of 41Cr4 steel - analysis of cracks grow in a plane strain state domination based on experimental results”
REVIEW No. 3 - note 4: 2. In Abstract and Summary sections, please remove the citation.
RESPONSE: All citations have been removed from the Abstract and the Summary sections.
REVIEW No. 3 - note 5.1: 3. In introduction section the author should provide more research that have been done in this field to provide sufficient background and relevant references. Sufficient information about the previous study findings should be added and compared with the present study so that the readers could follow the present study rationale and procedures.
REVIEW No. 3 - note 5.2 :4. Please explain more about the material 41Cr4steel. Why did the author choose this material and where this material is usually applied in the industry?
RESPONSE:. Appropriate corrections were added to the revised version of the paper. The research that had been carried out before was mentioned, reference was made to it, and a commentary on the genesis of the paper was added - where did the idea for researching the material come from. The section about the using the 41Cr4 steel in industry was added. New figure was added to manuscript, which presets the Everything was corrected as suggested by the Honorable Reviewer.
I assume that my comments and explanations have dispelled all doubts of the Honorable Reviewer. I believe that the corrections have made the manuscript more valuable and can be recommended for publication.

Round 2
Reviewer 1 Report
The manuscript can be accepted in the present form.
Reviewer 2 Report
Authors revised